# Contextual Latent World Models for Offline Meta Reinforcement Learning

## Abstract

Offline meta-reinforcement learning seeks to overcome the challenges of poor generalization and expensive data collection by leveraging datasets for related tasks. Context encoding is a prevalent approach, where an encoder maps transition histories to a task representation. In parallel, latent world models – which map observations into temporally consistent latent spaces – advanced self-supervised representation learning for planning and policy optimization. In this work, we unify these directions by introducing **contextual latent world models**: world models conditioned on the task representation and trained jointly with the context encoder. Coupling task inference with predictive modeling yields task representations that capture variation factors across tasks and empirically improves generalization to out-of-distribution tasks in diverse benchmarks, including MuJoCo, Contextual-DeepMind Control suite, and Meta-World.

## 1 Introduction

Reinforcement learning (RL) methods utilize predictive dynamics and reward functions in different ways. Model-based approaches attempt to increase sample efficiency with predictive models (Deisenroth & Rasmussen, 2011; Chua et al., 2018; Janner et al., 2019) to unroll trajectories into the future. On the other hand, model-free approaches use predictive models to improve value estimation (Feinberg et al., 2018) or to guide exploration (Stadie et al., 2015; Houthooft et al., 2016; Pathak et al., 2017; Achiam & Sastry, 2017; Pathak et al., 2019; Shyam et al., 2019; Scannell et al., 2024b) toward uncertain regions. Instead of learning predictive dynamics and reward functions directly in the observation/state space, latent world models map the observations to a (compact) latent vector and then learn a latent dynamics. Model-based approaches make use of latent world models for real-time planning (Hafner et al., 2019; Schrittwieser et al., 2020; Hansen et al., 2022; 2024; Scannell et al., 2025) or optimizing the policy by imagination (Hafner et al., 2020; 2021; 2025). Model-free approaches exploit latent world models for representation learning (Zhao et al., 2023; Fujimoto et al., 2023; Scannell et al., 2024a; Fujimoto et al., 2025). Utilizing latent world models in both model-based and model-free RL can improve sample efficiency and training stability considerably.

A key shortcoming of RL methods is limited generalization: a policy trained on one task typically cannot be directly applied to related tasks. Meta-RL aims to sidestep this issue by considering a distribution over a set of training tasks and learning a generalizable policy that can adapt within a few trials (Finn et al., 2017; Zintgraf et al., 2021; Beck et al., 2025). However, meta-RL is limited to simulation since direct interaction with a set of training tasks can be costly and even infeasible in the real world. Offline meta-RL (OMRL) attempts to overcome this issue by assuming that datasets exist for similar tasks and then leveraging the datasets to learn a generalizable policy. In context-based approaches (Li et al., 2020; Gao et al., 2024; Zhou et al., 2024; Li et al., 2024; Nakhaeinezhadfard et al., 2025; Wang et al., 2023; Zhang et al., 2025), a context encoder encodes a history of transitions to a latent vector called the task representation, and the agent (including policy and value functions) is conditioned on the task representation. The task representation serves as an implicit task identifier without requiring knowledge about the underlying task and the variation factors.

In this paper, we explore latent world models in the context-based OMRL setting. Our contributions are as follows:

C1 We present **C**ontextual **D**iscrete **C**odebook **W**orld **M**odels (C-DCWM), a novel OMRL method which is based on latent world models. More specifically, we extend discrete code-book world models (DCWM, Scannell et al., 2025) by conditioning the latent world model on the task representation and training the encoder and the world model jointly. We show that our approach of representation learning can improve generalization to unseen tasks.

C2 We compare different latent space formulations in world modeling and demonstrate the benefit of discretizing the latent space and training with a classification objective for the temporal consistency loss in the context-based OMRL setting.

C3 We evaluate task representation learning through the lens of disentanglement, showing that latent temporal consistency can better capture variation factors compared to predictive models, while including contrastive learning can enhance task distinguishability.

## 2 BACKGROUND

In this section, we review context-based offline meta-RL and introduce finite scalar quantization (FSQ), both of which are central to our method.

### 2.1 CONTEXT-BASED OFFLINE META REINFORCEMENT LEARNING

In offline meta-RL, there is a set of training tasks, each modeled as a Markov Decision Process (MDP), $\mathcal{M}_i = \langle \mathcal{S}, \mathcal{A}, R_i, P_i, \gamma, \rho_0 \rangle$, consisting of a shared state space $\mathcal{S}$, action space $\mathcal{A}$, discount factor $\gamma \in [0, 1]$, initial state distribution $\rho_0(s_0)$, a task-specific reward function $R_i : \mathcal{S} \times \mathcal{A} \to \mathbb{R}$, and task-specific transition dynamics $P_i(s_{t+1}|s_t, a_t)$. For each task represented as an MDP, there is a corresponding dataset $\mathcal{D}_i$. The objective is to train a meta-policy $\pi$ that can generalize to new tasks, *i.e.*, maximize the expected cumulative reward over the distribution of test tasks

$$J(\pi) = \mathbb{E}_{\mathcal{M}_i \sim p_{\text{test}}(\mathcal{M})} \left[ \mathbb{E}_{s_0 \sim \rho_0(s_0), s_{t+1} \sim P_i(\cdot|s_t, a_t), a_t \sim \pi(a_t|\cdot)} \left[ \sum_{t=0}^{T} \gamma^t R_i(s_t, a_t) \right] \right]. \quad (1)$$

Context-based methods use an encoder, called the context encoder, to implicitly infer the task $\mathcal{M}_i$ from limited samples collected by interacting with the environment. During meta training, a context encoder $E_\phi : \mathcal{S} \times \mathcal{A} \times \mathbb{R} \times \mathcal{S} \to \mathcal{Z}$ learns a mapping from transitions $\{(s_j, a_j, r_j, s'_j)\}$ to a task representation $z$. This task representation can be used *e.g.*, by the policy $\pi(a_i \mid s_i, z)$, Q-value function $Q(s_i, a_i, z)$, or a learned dynamics model $p(s_{t+1} \mid s_t, a_t, z)$ to adapt to the identified task.

### 2.2 FINITE SCALAR QUANTIZATION

The goal of quantization is to learn a codebook $\mathcal{C}$ whose elements provide a compressed represen-tation of the input data. Unlike vector quantization (VQ, Van Den Oord et al., 2017), which maps

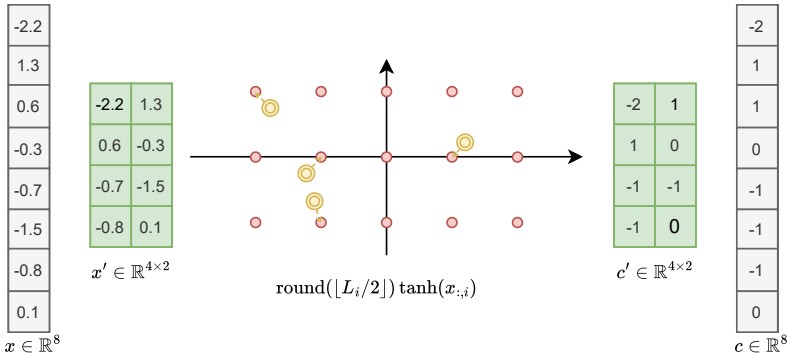

Figure 1: **Illustration of FSQ with two levels** $\mathcal{L} = [5, 3]$ The continuous vector $x$ is first reshaped into $|\mathcal{L}|$ channels, each channel $b_i$ is quantized to the nearest integer according to its resolution $L_i$, and the resulting quantized representation is then mapped back to the original dimensionality of $x$.

a continuous latent vector to the nearest codebook entry, finite scalar quantization (FSQ, Mentzer et al., 2024) divides the latent space into $b$ channels and quantizes each channel independently via bounded rounding, potentially with different resolutions. Concretely, a continuous latent vector $x \in \mathbb{R}^d$ is reshaped into $x' \in \mathbb{R}^{d' \times b}$, where $d' = d/b$ denotes the latent dimension per channel. Each latent dimension is quantized into an independent codebook, resulting in $d'$ codebooks in total. FSQ defines an ordered set of quantization levels $\mathcal{L} = [L_1, L_2, \ldots, L_b]$, where each $L_i$ specifies the resolution (number of quantized values) for channel $b_i$. Quantization is performed as $c'_{:,i} = \text{round}\left(\left\lfloor \frac{L_i}{2} \right\rfloor \tanh(x'_{:,i})\right)$, which maps each channel to $L_i$ discrete integer values. Consequently, each codebook over the $d'$ latent dimensions contains $|\mathcal{C}| = \prod_i L_i$ codes. The quantized vectors $c' \in \mathbb{R}^{d' \times b}$ are then reshaped back to $c \in \mathbb{R}^d$, preserving the dimensionality of the original latent vector $x$. Figure 1 illustrates this process. To enable gradient propagation through the non-differentiable rounding operation, FSQ employs the straight-through estimator (STE, Bengio, 2013). This approach produces a fixed grid partition in a lower-dimensional space, eliminating the need for the commitment and codebook losses typically used in VQ. As a result, FSQ yields efficient and stable discretization of the latent space.

## 3 METHOD

We start by providing a general overview, and then we detail our method for contextual world models and policy optimization. Fig. 2 provides a high-level illustration of C-DCWM.

**Overview**   C-DCWM has the following main components:

| | | |
|---|---|---|
| Context encoder: | $z = \mathbb{E}[E_\theta(s_t, a_t, r_t, s_{t+1})]$ | (2) |
| Observation encoder: | $x_t = F_\phi(s_t)$ | (3) |
| Quantization model: | $c_t = f(x_t)$ | (4) |
| Latent dynamics: | $\hat{c}_{t+1} \sim \text{Categorical}(p_1, ..., p_{|c|})$ where $p_i = D_\phi(\hat{c}_{t+1} = c_i \mid c_t, a_t, z)$ | (5) |
| Reward function: | $q = R_\phi(c_t, a_t, z)$ | (6) |
| Q-value function: | $q = Q_\psi(c_t, a_t, z)$ | (7) |
| Value function: | $v = V_\omega(c_t, z)$ | (8) |
| Policy: | $a_t \sim \pi_\eta(a_t \mid c_t, z)$ | (9) |

The latent world model in C-DCWM comprises a context encoder, an observation encoder, a quantization module, latent dynamics, and a reward function. The policy, value function, and Q-function are conditioned on both the discrete latent codes $c_t$ and the task representation $z$. This design is related to TCRL (Zhao et al., 2023), which demonstrates that converting the observation space of an RL agent using latent temporal consistency can enhance performance. On the other hand, our approach leverages latent temporal consistency within the offline meta-RL setting by jointly training the world model and the context encoder.

### 3.1 CONTEXTUAL DISCRETE CODEBOOK WORLD MODEL

In context-based OMRL methods, a context encoder maps transitions into a task representation $z$, which is then used to condition the agent and enable generalization to new tasks. Analogously, we extend discrete codebook world models (DCWM, Scannell et al., 2025) by conditioning them on task representations. Specifically, we condition the latent dynamics $D_\phi$ and reward function $R_\phi$ on $z$, while sharing the observation encoder $F_\phi$ and the quantization model across tasks. This design reflects the fact that tasks differ in dynamics and/or reward functions.

At each training iteration, a meta-batch of datasets is sampled, and from each dataset, a batch of transitions is drawn. Given a dataset $\mathcal{D}_i$ corresponding to training task $i$ (formulated as an MDP $\mathcal{M}_i$), the context encoder computes the task representation as

$$z^i = \mathbb{E}_{(s_t, a_t, r_t, s_{t+1}) \sim \mathcal{D}_i}[E_\phi(s_t, a_t, r_t, s_{t+1})]. \tag{10}$$

The observation encoder maps states from all tasks into continuous latent vectors according to Eq. (3), which are subsequently quantized using FSQ. Following DCWM, we use quantization levels

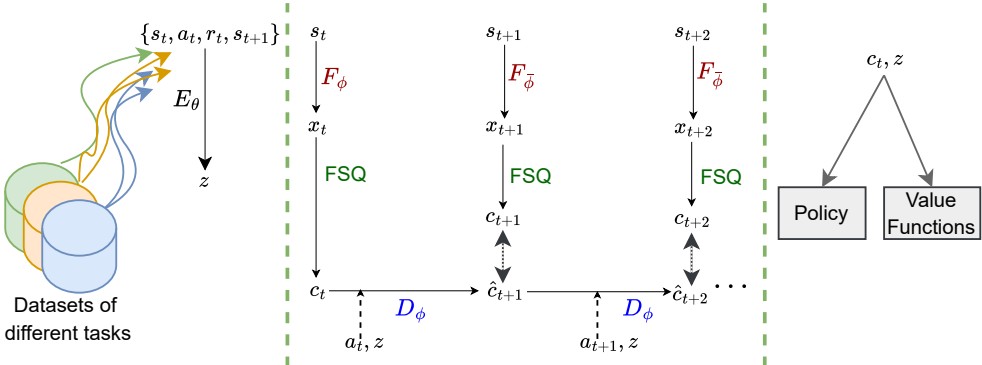

Figure 2: **Method Overview. Left:** Context encoder $E_\theta$ maps transitions of different tasks (from datasets during training and from previous interactions in testing) to a task representation, *i.e.* an implicit task identifier. **Middle:** During world model training, the observation encoder $F_\phi$ first maps observations $s_t$ to a latent vector $x_t$, which is then quantized – using FSQ – to a latent code $c_t$. The latent transition dynamics $D_\phi$ predicts the next latent quantized vectors $\hat{c}_{t+1:t+h}$ conditioned on the task representation $z$ computed by the context encoder. We use the cross-entropy loss (*i.e.* classification loss) between predictions from the dynamics model and predictions from the encoder at the next time step, to train the world model and context encoder jointly. **Right:** Policy optimization, the policy is optimized based on the quantized latent vectors $c_t$ while conditioned on the task representation $z$.

$\mathcal{L} = [5, 3]$, with codes normalized to lie in $[-1, 1]$. Given the discrete latent space, transition dynamics are modeled as categorical distributions over the next latent state. The latent space is divided into $d' = \frac{d}{|\mathcal{C}|}$ codebooks, each containing $|\mathcal{C}| = \prod_i L_i = 15$ codes. For each codebook, the latent dynamics outputs unnormalized logits over possible next codes $\hat{c}_{t+1}$ conditioned on the current code $c_t$, action $a_t$, and task representation $z$ according to Eq. (5). Probabilities for $|\mathcal{C}| = 15$ codes are obtained via a softmax normalization, yielding a potentially multimodal, stochastic transition distribution. To enable gradient-based training, we employ the straight-through Gumbel-Softmax estimator (Jang et al., 2017) for sampling. We jointly optimize the context encoder, observation encoder, latent dynamics, and reward function using backpropagation through time with the world model objective:

$$L_{\text{WM}}(\theta, \phi) = \sum_{h=0}^{H-1} \gamma^h \Big( \text{CE}(D_\phi(\hat{c}_{t+h}, a_{t+h}, z), c_{t+h+1}) + \|R_\phi(\hat{c}_{t+h}, a_{t+h}, z) - r_{t+h}\|_2^2 \Big) \quad (11)$$

$$\text{with } \underbrace{\hat{c}_0 = f(F_\phi(s_t))}_{\text{First latent state}}, \quad \underbrace{\hat{c}_{t+h+1} \sim D_\phi(\hat{c}_{t+h}, a_{t+h}, z)}_{\text{Stochastic latent dynamics}}, \quad \underbrace{c_{t+h} = \text{sg}(f(F_{\bar\phi}(s_{t+h})))}_{\text{Target latent code}}. \quad (12)$$

Here, $H$ denotes the multi-step prediction horizon, $\gamma$ the discount factor, $\bar\phi$ the exponential moving average of the observation encoder parameters, and CE the cross-entropy loss.

To ensure that task representations are discriminative, the context encoder is trained with a contrastive objective. Specifically, transitions from the same task should map to nearby representations, while those from different tasks should be further apart. We adopt the InfoNCE loss (van den Oord et al., 2019):

$$L_{\text{Contrastive}}(\theta) = -\sum_i \log \frac{S(z^i, \bar{z}^i)}{\sum_j S(z^i, \bar{z}^j)}, \quad (13)$$

where $\bar{z}^i = \lambda z^i + (1 - \lambda)\bar{z}^i$ is the moving average of task representations controlled by $\lambda$, and $S(z^i, z^j) = \exp\big(-\|z^i - z^j\|_2^2/\alpha\big)$ is an RBF kernel measuring similarity. This objective provides a lower bound on the mutual information between tasks and task representations, $I(z; M)$ (Zhang et al., 2024). Positive samples are obtained from the moving average of the same task representation, while negatives are drawn from other tasks. The moving average stabilizes training by smoothing updates. In practice, the context encoder is optimized with a combined objective:

$$L_{\text{Context Encoder}}(\theta) = L_{\text{WM}}(\theta) + \beta L_{\text{Contrastive}}(\theta),$$

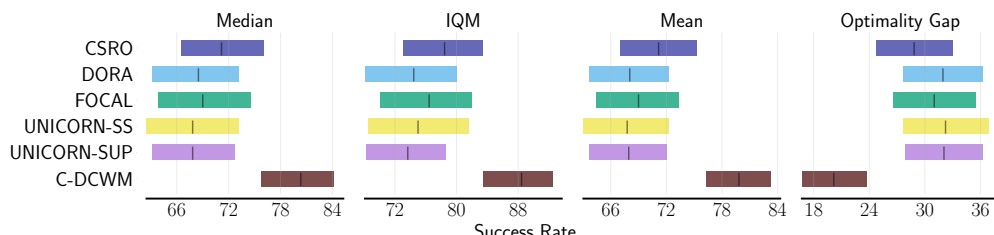

Figure 3: Few-shot in-distribution performance, including success rate (median, IQM, mean) and optimality gap (lower is better), on Meta-World benchmark (30 environments, each with 6 random seeds). **C-DCWM with the latent world model outperforms baselines**.

Table 1: Few-shot in-distribution performance on MuJoCo and Contextual-DMC benchmarks. **C-DCWM with the latent world model outperforms baselines**. Average returns over 6 random seeds, $\pm$ represents 95% confidence intervals. **Bold** indicates the highest value with statistical significance according to the t-test with p-value $< 0.05$.

| Environment | C-DCWM | CSRO | DORA | FOCAL | UNICORN-SS | UNICORN-SUP |
|---|---|---|---|---|---|---|
| Ant-dir | **863.1** $\pm$ 36.2 | 798.0 $\pm$ 39.3 | 596.5 $\pm$ 54.6 | 804.0 $\pm$ 35.0 | 812.9 $\pm$ 24.5 | 429.0 $\pm$ 30.6 |
| Cheetah-LS | **944.8** $\pm$ 4.9 | 831.2 $\pm$ 60.0 | 895.3 $\pm$ 24.1 | 852.2 $\pm$ 26.4 | 795.9 $\pm$ 39.4 | 832.0 $\pm$ 52.9 |
| Cheetah-speed | **751.2** $\pm$ 27.9 | 576.3 $\pm$ 78.2 | 547.0 $\pm$ 45.6 | 515.7 $\pm$ 62.7 | 554.3 $\pm$ 71.6 | 586.4 $\pm$ 40.7 |
| Finger-LS | **968.0** $\pm$ 5.5 | 869.2 $\pm$ 46.6 | 822.3 $\pm$ 47.5 | 880.8 $\pm$ 39.2 | 885.6 $\pm$ 14.1 | 753.2 $\pm$ 56.3 |
| Finger-speed | **967.4** $\pm$ 2.0 | 631.6 $\pm$ 56.1 | 441.2 $\pm$ 33.6 | 609.5 $\pm$ 25.0 | 515.9 $\pm$ 25.5 | 526.9 $\pm$ 36.8 |
| Hopper-mass | 590.6 $\pm$ 3.5 | 476.4 $\pm$ 68.7 | 563.3 $\pm$ 26.8 | 572.7 $\pm$ 13.0 | 540.9 $\pm$ 36.5 | 442.5 $\pm$ 119.2 |
| Walker-friction | 563.6 $\pm$ 33.5 | 521.8 $\pm$ 34.4 | 487.7 $\pm$ 27.8 | 532.3 $\pm$ 46.7 | 485.5 $\pm$ 57.8 | 539.1 $\pm$ 13.7 |
| Walker-LS | 934.6 $\pm$ 20.1 | 899.2 $\pm$ 41.8 | 862.5 $\pm$ 55.5 | 875.0 $\pm$ 51.5 | 880.7 $\pm$ 54.7 | 914.2 $\pm$ 23.0 |
| walker-speed | **835.6** $\pm$ 37.3 | 771.2 $\pm$ 20.0 | 390.9 $\pm$ 84.0 | 768.9 $\pm$ 30.1 | 730.7 $\pm$ 48.6 | 518.6 $\pm$ 55.3 |

where $\beta$ balances world model learning and contrastive task discrimination. The remaining components of the contextual DCWM are optimized solely with the world model objective in Eq. (11).

## 3.2 META POLICY OPTIMIZATION

Context-based OMRL extends offline RL by conditioning the value functions and policy on task representation, thereby enabling the learning of generalizable policies from datasets corresponding to different tasks. A central challenge in offline RL is out-of-distribution (OOD) action selection during temporal-difference (TD) learning. Actor-critic methods without regularization overestimate the value function while the policy is trained to optimize it. In principle, any offline RL algorithm can be employed to mitigate this issue. We adopt *Implicit Q-Learning* (IQL Kostrikov et al., 2022). IQL utilizes expectile regression in policy evaluation to predict an upper expectile of the TD targets in SARSA style without querying OOD actions. In our setting, we replace raw observations with quantized latent vectors from the world model. The value function then approximates an expectile with respect to only the action distribution $L_V(\omega) = \mathbb{L}_2^\tau(Q_{\bar{\psi}}(c_t, a_t, z) - V_\omega(c_t, z))$ where $\mathbb{L}_2^\tau(x) = (\tau - \mathbb{1}(x < 0))x^2$ is $\tau$ expectile regression and $\bar{\psi}$ is exponential moving average of $\psi$. The value function is then used in computing the target for training the Q-value function $L_Q(\psi) = \|r_t + V_\omega(c_t, z) - Q_\psi(c_t, a_t, z)\|_2^2$. The policy is optimized based on advantage weighted regression (Peng et al., 2019) $L_\pi(\eta) = -\log \pi_\eta(c_t, z) \exp(\mathcal{B}A(c_t, a_t, z))$ where $A(c_t, a_t, z) = Q_\psi(c_t, a_t, z) - V_\omega(c_t, z)$ is the advantage function and $\mathcal{B} \in [0, \infty)$ is the inverse temperature hyperparameter.

## 4 EXPERIMENTS

We evaluate C-DCWM on a set of multi-task environments from MuJoCo (Todorov et al., 2012), Contextual DeepMind Control (Contextual-DMC Tunyasuvunakool et al., 2020; Rezaei-Shoshtari et al., 2022), and Meta-World (Yu et al., 2020) benchmarks. Our experiments seek to answer the following research questions:

RQ1 Does C-DCWM's representation learning based on latent world models improve the performance of context-based OMRL agents in few-shot and zero-shot settings?

RQ2 How does C-DCWM generalize to out-of-distribution tasks and new environments compared to baselines?

RQ3 How does the latent-state space formulation *e.g.*, *(i)* classification loss, *(ii)* discrete codebook, and *(iii)* bounding the latent space, affect the performance.

RQ4 How do different objectives for training the context encoder, *e.g.*, *(i)* only contrastive objective, *(ii)* only world model objective, *(iii)* combination of them, affect the performance and task representation learning?

RQ5 How important is bounding the task representation for offline-meta RL performance *e.g.*, tanh vs $\ell_2$-normalization vs hypercube with FSQ?

**Experimental Setup:** We compare C-DCWM against the following baselines:

- **FOCAL** (Li et al., 2020): trains the context encoder using a distance metric objective, minimizing the squared $\ell_2$ distance between task representations from the same task and the inverse squared $\ell_2$ distance between those from different tasks.

- **CSRO** (Gao et al., 2024): extends FOCAL by additionally reducing context distribution shift through minimizing the CLUB (Cheng et al., 2020) upper bound on mutual information.

- **DORA** (Zhang et al., 2024): employs the InfoNCE loss, also used in C-DCWM, to train the context encoder. In contrast, C-DCWM augments this with the latent world model objective and introduces a discrete latent observation space.

- **UNICORN**: trains the context encoder via conditional predictive dynamics and reward functions with reconstruction. *UNICORN-SS* augments this with the FOCAL objective, while *UNICORN-SUP* relies solely on predictive models. These approaches share similarities with C-DCWM, but unlike them, C-DCWM leverages a discrete latent space.

To generate the datasets, we use Dropout Q-function (DroQ, Hiraoka et al., 2022) and we train separate agents for each task. Each dataset consists of trajectories collected from rolling out the agent at different phases of the training. Each dataset contains 1000 trajectories from a random policy to potentially an expert policy by training the DroQ agent up to 1M steps.

In the MuJoCO and the Contextual-DMC benchmarks, we sample 20 tasks for training, 10 tasks with the same distributions for variation factors for in-distribution testing, and 10 tasks with different distributions for variation factors for out-of-distribution testing. In the Meta-World benchmark, we use *Meta-RL (ML1, ML10, ML45)* settings. The *ML1* setting involves generalization to goal variation within a single environment, where each environment consists of 50 different tasks (with different goal/object positions). We select 40 tasks for training and 10 tasks for in-distribution testing. The *ML10* and *ML45* settings involve generalization to previously unseen environments without providing any prior information (*e.g.*, task IDs) and include randomized goals. They consist of 10 and 45 training environments, respectively, along with 5 unseen testing environments. Sec. A provides more details including hyperparameters (Table 5), hardware, and environments (Table 6).

## 4.1 GENERALIZATION TO NEW TASKS AND ENVIRONMENTS

During meta-testing, the agent is not provided with prior information about the current task or environment and must infer it based on its interaction experience. Context-based OMRL methods embed the collected experience in the task representation vector $z$, which is initially set to zero. At each time step $t$, the agent stores the interaction data $(s_t, a_t, r_t, s_{t+1})$, referred to as the context, and updates the task representation $z$ according to Eq. (2). As the agent gathers more interaction data, the task representation progressively captures the underlying task more accurately, enabling better generalization.

Table 1 and Fig. 3 summarize the few-shot performance of different methods on in-distribution tasks. C-DCWM outperforms the baselines in almost all the environments. Fig. 4 illustrates generalization to out-of-distribution tasks, where the in-distribution tasks are highlighted. C-DCWM can better generalize to out-of-distribution tasks, outperforming baselines while demonstrating more consistent

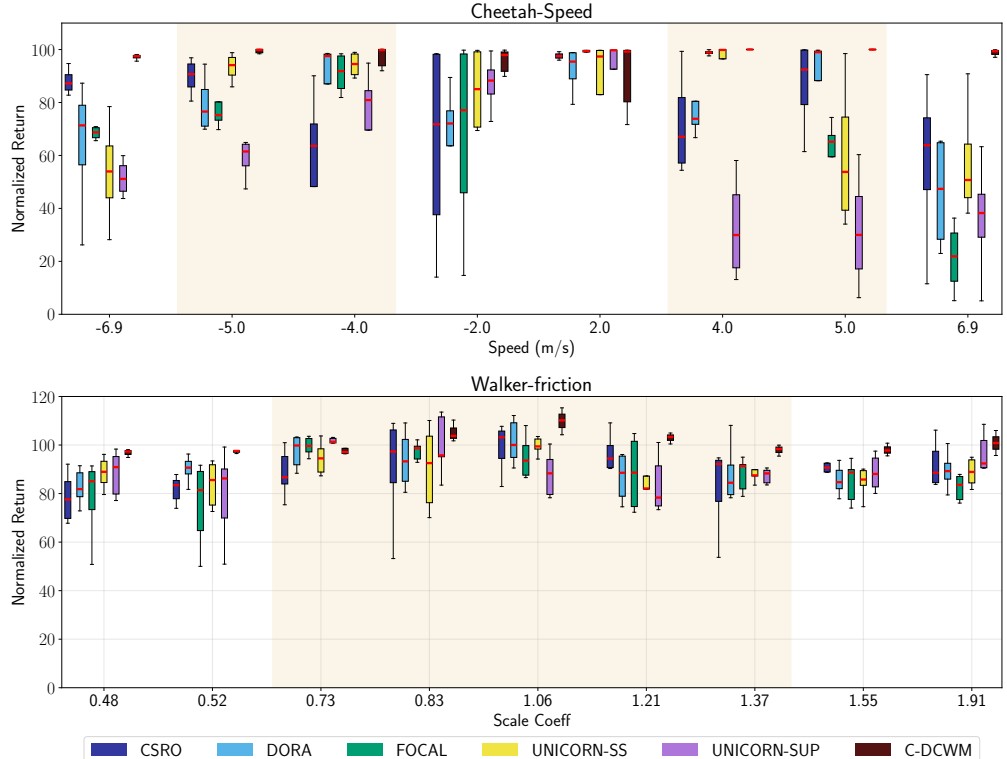

Figure 4: **C-DCWM outperforms baselines on both in-distribution (highlighted regions) and out-of-distribution tasks**, demonstrating superior generalization. Boxes represent the interquartile range with the median.

performance along different variation factors. These results suggest that jointly training the context encoder with latent temporal consistency and contrastive objectives, while leveraging a discrete latent world model, improves both in-distribution performance and generalization.

Table 2 summarizes the few-shot performance on new environments under the *ML10* and *ML45* setting in the Meta-World benchmark. C-DCWM outperforms baselines on both the training and testing environments in both settings. The performance on the testing environments is slightly better on the *ML45* setting, where a more diverse set of environments is considered for training. However, there is a significant performance drop when generalizing to entirely new environments, especially compared to generalization to new tasks within a single environment (*e.g.*, different goal or object positions).

Sec. B.1 provides few-shot results for all Meta-World environments in Table 7 and zero-shot results for MuJoCo and the Contextual-DMC benchmarks for both in-distribution tasks (Table 8) and out-of-distribution tasks (Table 9). Sec. B.3 investigates the impact of contrastive and latent temporal consistency objectives on generalization to both in-distribution (Table 12) and out-of-distribution (Table 13) tasks.

## 4.2 COMPARISON OF DIFFERENT LATENT SPACES

Different RL methods formulate the latent space in different ways; TD-MPC (Hansen et al., 2022) employs a continuous latent space with a mean squared error (MSE) loss for enforcing temporal consistency, while TCRL (Zhao et al., 2023) replaces this with a cosine similarity objective. TD-MPC2 (Hansen et al., 2024) further constrains the latent space using a *SimNorm* operation. iQRL (Scannell et al., 2024a) discretizes and bounds the latent space with FSQ while maximizing cosine similarity, and DC-MPC (Scannell et al., 2025) formulates latent temporal consistency as a classification task using a cross-entropy loss. We investigate the impact of these alternative formulations in our setting,

Table 2: **Generalization to new environments:** C-DCWM demonstrates better generalization to unseen environments. Increasing the number of training environments can improve generalization to testing environments. Average success rate over 6 random seeds, $\pm$ represents 95% confidence intervals. **Bold** indicates the highest value with statistical significance according to the t-test with p-value $< 0.05$.

| Setting | C-DCWM | CSRO | DORA | FOCAL | UNICORN-SS | UNICORN-SUP |
|---|---|---|---|---|---|---|
| ML10-Train | **83.5** $\pm$ 6.1 | 43.3 $\pm$ 7.6 | 49.6 $\pm$ 3.2 | 43.3 $\pm$ 6.5 | 43.3 $\pm$ 3.0 | 25.4 $\pm$ 3.7 |
| ML10-Test | **15.0** $\pm$ 3.7 | 4.2 $\pm$ 4.1 | 1.7 $\pm$ 2.1 | 5.0 $\pm$ 5.1 | 4.2 $\pm$ 4.7 | 2.5 $\pm$ 3.3 |
| ML45-Train | **58.7** $\pm$ 2.9 | 32.8 $\pm$ 2.4 | 32.6 $\pm$ 1.3 | 32.2 $\pm$ 2.9 | 30.7 $\pm$ 1.7 | 18.3 $\pm$ 1.2 |
| ML45-Test | **24.0** $\pm$ 10.0 | 8.3 $\pm$ 6.0 | 6.7 $\pm$ 6.5 | 3.3 $\pm$ 4.1 | 8.3 $\pm$ 10.6 | 8.3 $\pm$ 9.4 |

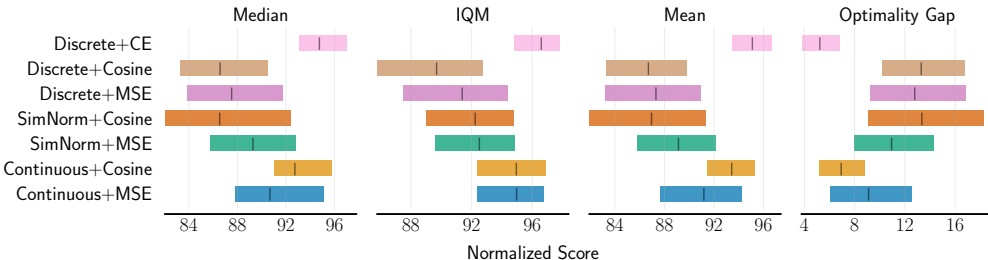

Figure 5: Normalized returns (median, IQM, mean) and optimality gap (lower is better) for different world modeling methods, evaluated on 9 environments with 6 random seeds per environment. **The main advantage of discretizing the latent space is due to classification** loss (cross entropy). Bounding or discretizing the latent space alone does not improve performance.

with results summarized in Fig. 5 for MuJoCo and Contextual-DMC benchmarks. Our findings indicate that bounding or discretizing the latent space, as well as replacing MSE with cosine similarity, does not yield performance gains. However, framing latent temporal consistency as a classification problem, rather than regression, can enhance performance. Similarly, Farebrother et al. (2024) demonstrates the advantages of classification compared to regression in training value functions.

## 4.3 INVESTIGATING TASK REPRESENTATION

Previous OMRL methods have investigated task representation learning using t-SNE (van der Maaten & Hinton, 2008), which maps task representations for different tasks to a lower-dimensional space (typically two-dimensional) for visualization to assess task distinguishability. However, this approach has limitations: it evaluates the context encoder's ability to distinguish tasks solely through visualization and does not account for the relationship between true variation factors and the task representations, and is sensitive to hyperparameters and initialization (Wang et al., 2021).

To address these limitations, we propose using disentanglement metrics, which quantify the extent to which latent vectors isolate individual variation factors, with each dimension ideally capturing only one factor. We employ DCI (Eastwood & Williams, 2018) metrics, which is based on regression models with an importance matrix, and InfoMEC (Hsu et al., 2023) metrics, which is based on normalized mutual information, to evaluate task representation learning across different objectives. The DCI metrics comprise **disentanglement**, which measures the degree to which each dimension captures a single variation factor; **completeness**, which assesses the extent to which each variation factor is modeled by a single dimension; and **informativeness**, which quantifies the information captured by the latent vector based on prediction error. The InfoMEC metrics include **modularity**, which evaluates the separation of variation factors into disjoint sets of latent dimensions; **explicitness**, which measures the simplicity with which latent vectors encode each variation factor; and **compactness**, which assesses the degree to which each dimension encodes information about disjoint sets of variation factors. When the latent vector dimensionality exceeds the number of variation factors, achieving perfect modularity and perfect compactness simultaneously is impossible (Hsu et al., 2023), and modularity should be prioritized.

Table 3: Disentanglement metrics (DCI, InfoMEC) for the Cheetah-length-speed (Ls) environment. **Latent world models disentangle the variation factors more effectively**, while **contrastive learning enhances task distinguishability**, reflected in informativeness and explicitness. *WM* denotes training the context encoder solely with the world model objective (Eq. (11)); *FOCAL* and *InfoNCE* represent two contrastive objectives, and *UNICORN-SUP* indicates training with reconstruction (decoder). Average metrics over 6 random seeds, $\pm$ represents $95\%$ confidence intervals.

| | Disentanglement | Completeness | Informativeness | Modularity | Explicitness | Compactness |
|---|---|---|---|---|---|---|
| FOCAL | $0.33 \pm 0.11$ | $0.25 \pm 0.08$ | $0.82 \pm 0.04$ | $0.77 \pm 0.04$ | $0.75 \pm 0.05$ | $0.20 \pm 0.05$ |
| CSRO | $0.31 \pm 0.05$ | $0.32 \pm 0.05$ | $0.83 \pm 0.04$ | $0.75 \pm 0.03$ | $0.78 \pm 0.03$ | $0.21 \pm 0.04$ |
| DORA (InfoNCE) | $0.24 \pm 0.04$ | $0.29 \pm 0.05$ | $0.75 \pm 0.04$ | $0.70 \pm 0.04$ | $0.78 \pm 0.03$ | $0.13 \pm 0.08$ |
| UNICORN-SS | $0.32 \pm 0.06$ | $0.29 \pm 0.09$ | $0.83 \pm 0.04$ | $0.76 \pm 0.01$ | $0.76 \pm 0.04$ | $0.23 \pm 0.05$ |
| UNICORN-SUP | $0.36 \pm 0.09$ | $0.23 \pm 0.06$ | $0.54 \pm 0.06$ | $0.76 \pm 0.05$ | $0.71 \pm 0.02$ | $0.24 \pm 0.07$ |
| WM | $0.45 \pm 0.06$ | $0.49 \pm 0.07$ | $0.81 \pm 0.03$ | $0.70 \pm 0.02$ | $0.82 \pm 0.02$ | $0.23 \pm 0.07$ |
| WM+FOCAL | $0.42 \pm 0.05$ | $0.47 \pm 0.06$ | $0.87 \pm 0.02$ | $0.71 \pm 0.03$ | $0.86 \pm 0.02$ | $0.24 \pm 0.06$ |
| WM+InfoNCE (ours) | $0.50 \pm 0.05$ | $0.49 \pm 0.05$ | $0.89 \pm 0.02$ | $0.74 \pm 0.03$ | $0.87 \pm 0.01$ | $0.26 \pm 0.05$ |

Table 4: **Bounding the task representation enables better generalization in certain environments**, though discretizing with FSQ yields no advantages. Average returns and success rates over 6 random seeds, $\pm$ represents $95\%$ confidence intervals.

| Environment | Identity | $\ell_2$-Norm | FSQ | Tanh |
|---|---|---|---|---|
| Ant-dir | $452.7 \pm 121.3$ | $838.0 \pm 72.9$ | $866.0 \pm 46.2$ | $863.1 \pm 36.2$ |
| Cheetah-LS | $933.8 \pm 10.5$ | $932.3 \pm 16.7$ | $938.3 \pm 5.8$ | $944.8 \pm 4.9$ |
| Cheetah-speed | $670.3 \pm 75.1$ | $778.0 \pm 27.8$ | $730.7 \pm 30.2$ | $764.1 \pm 39.2$ |
| Finger-LS | $962.2 \pm 3.8$ | $956.6 \pm 11.4$ | $969.6 \pm 5.4$ | $968.0 \pm 5.5$ |
| Finger-speed | $715.4 \pm 133.6$ | $855.1 \pm 83.8$ | $950.4 \pm 30.4$ | $967.4 \pm 2.0$ |
| Walker-LS | $922.7 \pm 20.0$ | $924.7 \pm 15.7$ | $932.4 \pm 42.9$ | $934.6 \pm 20.1$ |
| Button-press | $98.3 \pm 3.3$ | $100.0 \pm 0.0$ | $98.3 \pm 3.3$ | $100.0 \pm 0.0$ |
| Coffee-button | $100.0 \pm 0.0$ | $100.0 \pm 0.0$ | $100.0 \pm 0.0$ | $100.0 \pm 0.0$ |
| Dial-turn | $96.7 \pm 4.1$ | $95.0 \pm 4.4$ | $78.0 \pm 33.6$ | $96.7 \pm 4.1$ |
| Door-open | $95.0 \pm 6.7$ | $98.3 \pm 3.3$ | $100.0 \pm 0.0$ | $100.0 \pm 0.0$ |
| Door-unlock | $98.3 \pm 3.3$ | $96.7 \pm 4.1$ | $98.3 \pm 3.3$ | $100.0 \pm 0.0$ |

Table 3 summarizes the disentanglement scores for different methods in the Cheetah-LS environment, which features two variation factors: desired speed and torso length. We select 10 in-distribution and 10 out-of-distribution tasks, collecting 1000 samples per task while updating the task representation (akin to a few-shot setting). Here, *WM* denotes training the context encoder solely with the world model objective (Eq. (11)); *FOCAL* and *InfoNCE* represent two contrastive objectives; and *UNICORN-SUP* indicates training with a conditional predictive model (decoder). Comparing *WM* and *UNICORN-SUP* shows that leveraging the latent world model, rather than a predictive model, improves disentanglement metrics. Incorporating contrastive learning objectives enhances task distinguishability (as reflected in informativeness and explicitness). Sec. B.2 provides disentanglement scores for additional environments.

## 4.4 BOUNDING THE TASK REPRESENTATION

In this section, we evaluate how bounding the task representation $z$ affects the generalization. By default, C-DCWM utilize Tanh as the activation function for the context encoder, consistent with prior OMRL methods. Table 4 summarize the results. We compare unbounded representations (Identity), $\ell_2$-normalization, FSQ, and Tanh for bounding the latent space. Since we utilize FSQ in our latent world model, we investigate whether discretizing the task representation with FSQ influences generalization. The results indicate that bounding the task representation significantly enhances generalization in certain environments (*e.g.*, Ant-dir, Cheetah-speed). Discretizing the task representation with FSQ yields no advantages over Tanh, and $\ell_2$-normalization is less robust across different environments.

## 5 RELATED WORK

**Latent World Models** Ha & Schmidhuber (2018) introduced world models, wherein a variational autoencoder compresses the observation space, and a recurrent neural network models the dynam-

ics in the latent space. PlaNet (Hafner et al., 2019) employs a recurrent state-space model (RSSM; Doerr et al., 2018), jointly training the encoder, latent dynamics, and reward function by maximizing the evidence lower bound (via reconstruction) and performing decision-time planning. Dreamer (Hafner et al., 2020) optimizes the policy using value functions in imagined trajectories generated by the latent world model. Subsequent versions (Hafner et al., 2021; 2025) incorporate discrete latent spaces (in the form of one-hot encoding) trained with classification objectives, yielding significant performance enhancements. In contrast, TD-MPC (Hansen et al., 2022; 2024) relies on latent temporal consistency within a continuous latent space, eschewing reconstruction. DC-MPC (Scannell et al., 2025) discretizes the latent space and employs classification for temporal consistency, demonstrating superior performance in continuous control. Model-free methods such as TCRL (Zhao et al., 2023), TD7 (Fujimoto et al., 2023), and MR.Q (Fujimoto et al., 2025) leverage latent world models to modify or augment representations for policies and value functions based on temporal consistency.

**Context-based Offline Meta-RL** Offline meta-RL (OMRL) methods seek to enable policies to generalize to unseen tasks within a few trials, leveraging datasets from a distribution of related tasks. Contrastive learning has been utilized to train context encoders (Li et al., 2020; Yuan & Lu, 2022). However, contrastive learning fails to address context distribution shifts, which arise from discrepancies between the distributions of context samples during training and testing, due to differences between the learned policy and the behavior policies that collected the datasets. CSRO (Gao et al., 2024) mitigates this shift by approximately minimizing the mutual information between task representations and behavior policies. ER-TRL (Nakhaeinezhadfard et al., 2025) reformulates this mutual information in entropy terms and shows that maximizing the entropy of a meta-behavior policy can alleviate the distribution shift. UNICORN (Li et al., 2024) addresses this issue using predictive models, demonstrating that reconstructing the next state and reward via conditional dynamics and reward predictors encourages the context encoder to encode task-relevant information. C-DCWM, in contrast, employs latent world models.

## 6 CONCLUSION

This paper presents a novel approach to offline meta-RL, **contextual latent world models**, wherein world models are conditioned on the task representation. We train the latent world model and context encoder jointly with latent temporal consistency and contrastive learning. We compare various latent space formulations and demonstrate that a discrete latent space with classification-based temporal consistency yields superior results. We then use the discrete latent state and task representation for policy optimization. Empirical results indicate that this representation learning paradigm more effectively captures underlying variation factors and exhibits enhanced generalization.

**Limitations** Extending our framework to accommodate different state and action spaces across tasks is a promising direction for future research. Furthermore, we evaluate C-DCWM in environments with only state information; in the future, the observation encoder architecture could be modified to support visual observations.

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

APPENDICES

In this appendix, we provide details of our method in Sec. A and additional results in Sec. B.

LARGE LANGUAGE MODELS

We use large language models (LLMs) to assist with paper writing, including proofreading for typos and grammar errors. We also employ LLMs to generate scripts for visualizing results and creating figures.

# A    IMPLEMENTATION DETAILS

We implemented C-DCWM with PyTorch (Paszke et al., 2019) and used the AdamW optimizer (Loshchilov & Hutter, 2019) for training the world model and the Adam optimizer (Kingma & Ba, 2017) for the other models. All neural networks are implemented as MLPs where each intermediate linear layer is followed by Layer Normalization (Ba et al., 2016) and Mish activation function (Misra, 2019). Below we summarize the architecture of C-DCWM for the Cheetah-LS environment.

```
Context Encoder: Mlp(
  (net): Sequential(
    (0): NormedLinear(in_features=41, out_features=256, bias=True, act=Mish)
    (1): NormedLinear(in_features=256, out_features=256, bias=True, act=Mish)
    (2): NormedLinear(in_features=256, out_features=256, bias=True, act=Mish)
    (3): Linear(in_features=256, out_features=10, bias=True)
  )
)
World Model: ContextualWorldModel(
  (Fsq): FSQ(levels=[5, 3])
  (Encoder): Mlp(
    (net): Sequential(
      (0): NormedLinear(in_features=17, out_features=512, bias=True, act=Mish)
      (1): Linear(in_features=512, out_features=1024, bias=True)
    )
  )
  (Encoder_tar): Mlp(
    (net): Sequential(
      (0): NormedLinear(in_features=17, out_features=512, bias=True, act=Mish)
      (1): Linear(in_features=512, out_features=1024, bias=True)
    )
  )
  (Trans): Mlp(
    (net): Sequential(
      (0): NormedLinear(in_features=1040, out_features=512, bias=True, act=Mish)
      (1): NormedLinear(in_features=512, out_features=512, bias=True, act=Mish)
      (2): Linear(in_features=512, out_features=7680, bias=True)
    )
  )
  (Reward): Vectorized [Mlp(
    (net): Sequential(
      (0): NormedLinear(in_features=1040, out_features=512, bias=True, act=Mish)
      (1): NormedLinear(in_features=512, out_features=512, bias=True, act=Mish)
      (2): Linear(in_features=512, out_features=1, bias=True)
    )
  )]
)
Policy: Mlp(
  (net): Sequential(
    (0): NormedLinear(in_features=1034, out_features=256, bias=True, act=Mish)
    (1): NormedLinear(in_features=256, out_features=256, bias=True, act=Mish)
    (2): Linear(in_features=256, out_features=12, bias=True)
  )
)
Q-Functions: Vectorized [Mlp(
  (net): Sequential(
    (0): NormedLinear(in_features=1040, out_features=256, bias=True, act=Mish)
    (1): NormedLinear(in_features=256, out_features=256, bias=True, act=Mish)
    (2): Linear(in_features=256, out_features=1, bias=True)
  )
), Mlp(
  (net): Sequential(
    (0): NormedLinear(in_features=1040, out_features=256, bias=True, act=Mish)
    (1): NormedLinear(in_features=256, out_features=256, bias=True, act=Mish)
```

```
    (2): Linear(in_features=256, out_features=1, bias=True)
  )
)]
Value Function: Mlp(
  (net): Sequential(
    (0): NormedLinear(in_features=1034, out_features=256, bias=True, act=Mish)
    (1): NormedLinear(in_features=256, out_features=256, bias=True, act=Mish)
    (2): Linear(in_features=256, out_features=1, bias=True)
  )
)
Learnable parameters: 7.55 M
```

**Hardware**   We used AMD Instinct MI250X GPUs to run our experiments. All experiments have been run on a single GPU with 2 CPU workers and 32GB of RAM.

**Hyperparameters**   Table 5 illustrates the hyperparameters for our experiments. We use the same hyperparameters for all of the experiments. For a fair comparison, we use the same network architecture for all the baselines.

Table 5: Hyperparameters of our method C-DCWM.

| HYPERPARAMETER | VALUE | DESCRIPTION |
|---|---|---|
| **DATA COLLECTION** | | |
| TRAIN STEPS | $10^6$ | |
| RANDOM STEPS | $5 \times 10^4$ | NUM. RANDOM STEPS AT START |
| NUM. EVAL EPISODES | 50 | NUM. TRAJECTORIES IN EVALUATION |
| EVAL. EVERY STEPS | $5 \times 10^4$ | |
| POLICY MLP DIMS | $[512, 512]$ | |
| VALUE FUNCTION MLP DIMS | $[512, 512]$ | |
| DROPOUT RATIO | 0.1 | |
| LEARNING RATE | $10^{-4}$ | VALUE FUNCTION, ENTROPY COEFF |
| | $3 \times 10^{-4}$ | POLICY |
| TARGET ENTROPY | $-\|\mathcal{A}\|_1$ | |
| BATCH SIZE | 1024 | |
| DISCOUNT FACTOR $\gamma$ | 0.99 | |
| MOMENTUM COEF | 0.005 | SOFT UPDATE TARGET NETWORKS |
| **CONTEXTUAL LATENT WORLD MODEL** | | |
| OBSERVATION ENCODER MLP DIMS | $[512]$ | |
| CONTEXT ENCODER MLP DIMS | $[256, 256]$ | |
| LATENT DYNAMICS AND REWARD MLP DIMS | $[512, 512]$ | |
| TASK REPRESENTATION DIM | 5 | |
| LATENT DIM | 1024 | |
| FSQ LEVELS | $[5, 3]$ | |
| CONSISTENCY COEFF | 1.0 | |
| REWARD COEFF | 1.0 | |
| DISCOUNT FACTOR $\gamma$ | 0.99 | |
| TRAINING HORIZON $H$ | 5 | |
| LEARNING RATE | $10^{-4}$ | |
| CONTRASTIVE OBJ WEIGHT $\beta$ | 1.0 | |
| MOMENTUM COEF | 0.005 | SOFT UPDATE TARGET ENCODER |
| **OFFLINE META-RL** | | |
| META BATCH SIZE | 16 | |
| BATCH SIZE | 256 | |
| NUM. TRAIN TASK | 20 | MUJOCO & DMC |
| | 40 | METAWORLD |
| NUM. EVAL TASK | 10 | |
| NUM. OOD TASK | 10 | MUJOCO & DMC |
| | 0 | METAWORLD |
| CONTEXT SIZE | 256 | |
| BUFFER SIZE | $2 \times 10^5$ | FOR EACH TASK |
| DISCOUNT FACTOR $\gamma$ | 0.99 | |
| (Q-)VALUE FUNCTION MLP DIMS | $[256, 256]$ | |
| NUM. Q FUNCTIONS | 2 | |
| POLICY MLP DIMS | $[256, 256]$ | |
| LEARNING RATE | $3 \times 10^{-4}$ | POLICY AND (Q-)VALUE FUNCTIONS |
| EXPECTILE REGRESSION $\tau$ | 0.8 | |
| INVERSE TEMPERATURE $\mathcal{B}$ | 3.0 | IN POLICY OPTIMIZATION |
| MOMENTUM COEF. | 0.005 | SOFT UPDATE TARGET NETWORKS |
| NUM. TEST TRAJECTORIES | 3 | K FEW-SHOT |

**Environments**   We evaluated C-DCWM on 3 MuJoCo (Todorov et al., 2012) environments, 6 Contextual DeepMind Control (Tunyasuvunakool et al., 2020; Rezaei-Shoshtari et al., 2022) envi-

ronments, and 50 Meta-World ML1 Beck et al. (2025) environments. Table 6 provides details of the environments we used, including the dimensionality of the observation and action space, and the distribution of variation factors

- **Ant-direction:** an ant (quadruped) robot moving in different desired directions in different tasks.

- **Hopper-mass:** a hopper (one-legged robot) must move as fast as it can, while the mass is different for different tasks.

- **Walker-friction:** a walker (bi-legged) robot must move as fast as it can, while the friction coefficient is different for different tasks.

- **Cheetah-speed:** a cheetah robot moving forwards/backwards with different desired speeds in different tasks.

- **Finger-speed:** a planar finger robot rotating a body on an unactuated hinge with different desired angular speeds (both directions) for different tasks.

- **Walker-speed:** a walker robot moving forwards/backwards with different desired speeds in different tasks.

- **Cheetah-length-speed (LS):** a cheetah robot moving forwards, where torso length (change in morphology) and/or desired speed are different for each task.

- **Finger-length-speed (LS):** a planar finger robot rotating a body on an unactuated hinge, while the length of the link and/or desired angular speed differ in each task.

- **Walker-length-speed (LS):** a walker robot moving forwards, where torso length (change in morphology) and/or desired speed are different for each task.

- **Meta-World ML1**: consists of 50 robotic manipulation environments featuring a Sawyer arm with various everyday objects. Each environment consists of 50 different tasks where the position of objects and goals is different for each task.

- **Meta-World ML10:** evaluates generalization to new environments. Similar to the ML1 setting, it consists of robotic manipulation environments, where 10 environments are used for training and 5 environments are reserved for testing generalization capabilities. The testing environments share structural similarities with the training environments. During testing, no prior information about the environment (such as environment ID) is provided, and agents must identify and adapt to the environment solely based on interaction data.

- **Meta-World ML45:** evaluates generalization to new environments, similar to the ML10 setting, but with a larger and more diverse set of 45 training environments.

Table 6: Environment used for evaluation of different methods.

| ENVIRONMENT | OBS DIM | ACTION DIM | ID VARIATION FACTORS | OOD VARIATION FACTORS |
|---|---|---|---|---|
| ANT-DIRECTION | 29 | 8 | $\theta \sim [-\pi, \pi]$ | $\theta \sim [-1.5\pi, -\pi] \cup [\pi, 1.5\pi]$ |
| HOPPER-MASS | 11 | 3 | $\log f_m \sim [-1.5, 1.5]$ | $\log f_m \sim [-2, -1.5] \cup [1.5, 2]$ |
| WALKER-FRICTION | 17 | 6 | $\log f_f \sim [-1.5, 1.5]$ | $\log f_f \sim [-2, -1.5] \cup [1.5, 2]$ |
| CHEETAH-SPEED | 17 | 6 | $v \sim [-10, -6] \cup [-2, 2] \cup [6, 10]$ | $v \sim [-6, -2] \cup [2, 6]$ |
| FINGER-SPEED | 17 | 6 | $v \sim [-15, -9] \cup [-3, 3] \cup [9, 15]$ | $v \sim [-9, -3] \cup [3, 9]$ |
| WALKER-SPEED | 24 | 6 | $v \sim [-5, -3] \cup [-1, 1] \cup [3, 5]$ | $v \sim [-3, -1] \cup [1, 3]$ |
| CHEETAH-LENGTH-SPEED | 17 | 6 | $v \sim [3, 8]$ $L \sim [0.4, 0.6]$ | $v \in \{1, 2, 9, 10\}$ $L \in \{0.3, 0.35, 0.65, 0.7\}$ |
| FINGER-LENGTH-SPEED | 9 | 2 | $v \sim [5, 10]$ $L \sim [0.15, 0.25]$ | $v \in \{3, 4, 11, 12\}$ $L \in \{0.1, 0.12, 0.27, 0.3\}$ |
| WALKER-LENGTH-SPEED | 24 | 6 | $v \sim [2, 4.5]$ $L \sim [0.2, 0.4]$ | $v \in \{1, 1.5, 5, 5.5\}$ $L \in \{0.1, 0.15, 0.45, 0.5\}$ |
| META-WORLD | 39 | 4 | | |

**Open-source code**  For full details of the implementation, model architectures, and training, please check the code, which is available in the submitted supplementary material and will be made public upon acceptance to guarantee reproducibility.

## B FURTHERE RESULTS

### B.1 GENERALIZATION TO NEW TASKS

Table 7 summarizes the results across all environments in the Meta-World ML1 benchmark. We collected datasets for these environments at varying difficulty levels using the same RL algorithm (DroQ) with identical hyperparameters. In certain environments, such as Assembly, the datasets lack successful trajectories. Consequently, various OMRL methods fail to learn the corresponding tasks. In the majority of environments, C-DCWM outperforms the baselines, achieving higher success rates during few-shot adaptation.

Table 7: Few-shot in-distribution performance on all environments in MetaWorld benchmarks. Average success rate over 6 random seeds, $\pm$ represents 95% confidence intervals. We use DroQ with the same hyperparameters to collect the datasets for all environments, resulting in a lack of successful trajectories in the datasets for some environments. **Bold** indicates the highest value with statistical significance according to the t-test with p-value $< 0.05$.

| Environment | C-DCWM | CSRO | DORA | FOCAL | UNICORN-SS | UNICORN-SUP |
|---|---|---|---|---|---|---|
| Assembly | $0.0 \pm 0.0$ | $0.0 \pm 0.0$ | $0.0 \pm 0.0$ | $0.0 \pm 0.0$ | $0.0 \pm 0.0$ | $0.0 \pm 0.0$ |
| Basketball | $\mathbf{8.3} \pm 6.0$ | $0.0 \pm 0.0$ | $0.0 \pm 0.0$ | $0.0 \pm 0.0$ | $0.0 \pm 0.0$ | $0.0 \pm 0.0$ |
| Bin-picking | $1.7 \pm 3.3$ | $3.3 \pm 4.1$ | $0.0 \pm 0.0$ | $0.0 \pm 0.0$ | $0.0 \pm 0.0$ | $0.0 \pm 0.0$ |
| Box-close | $\mathbf{26.7} \pm 9.7$ | $11.7 \pm 6.0$ | $1.7 \pm 3.3$ | $6.7 \pm 6.5$ | $3.3 \pm 4.1$ | $11.7 \pm 9.4$ |
| Button-press | $100.0 \pm 0.0$ | $95.0 \pm 6.7$ | $90.0 \pm 5.1$ | $98.3 \pm 3.3$ | $91.7 \pm 6.0$ | $90.0 \pm 7.2$ |
| Button-press-topdown | $83.3 \pm 33.1$ | $68.3 \pm 9.4$ | $75.0 \pm 18.8$ | $78.3 \pm 6.0$ | $83.3 \pm 6.5$ | $68.3 \pm 7.9$ |
| Button-press-topdown-wall | $\mathbf{90.0} \pm 5.1$ | $55.0 \pm 14.1$ | $36.7 \pm 8.3$ | $35.0 \pm 15.0$ | $35.0 \pm 11.0$ | $31.7 \pm 11.8$ |
| Button-press-wall | $96.7 \pm 4.1$ | $83.3 \pm 4.1$ | $86.7 \pm 4.1$ | $91.7 \pm 7.9$ | $91.7 \pm 6.0$ | $81.7 \pm 11.8$ |
| Coffee-button | $100.0 \pm 0.0$ | $98.3 \pm 3.3$ | $96.7 \pm 4.1$ | $95.0 \pm 6.7$ | $100.0 \pm 0.0$ | $100.0 \pm 0.0$ |
| Coffee-pull | $3.3 \pm 4.1$ | $1.7 \pm 3.3$ | $0.0 \pm 0.0$ | $1.7 \pm 3.3$ | $0.0 \pm 0.0$ | $3.3 \pm 4.1$ |
| Coffee-push | $25.0 \pm 8.4$ | $18.3 \pm 13.8$ | $13.3 \pm 8.3$ | $13.3 \pm 12.0$ | $18.3 \pm 7.9$ | $18.3 \pm 6.0$ |
| Dial-turn | $88.3 \pm 9.4$ | $93.3 \pm 6.5$ | $90.0 \pm 10.1$ | $90.0 \pm 8.8$ | $86.7 \pm 8.3$ | $65.0 \pm 16.6$ |
| Disassemble | $25.0 \pm 8.4$ | $16.7 \pm 8.3$ | $8.3 \pm 6.0$ | $5.0 \pm 6.7$ | $18.3 \pm 10.6$ | $6.7 \pm 6.5$ |
| Door-close | $100.0 \pm 0.0$ | $100.0 \pm 0.0$ | $100.0 \pm 0.0$ | $100.0 \pm 0.0$ | $100.0 \pm 0.0$ | $96.7 \pm 4.1$ |
| Door-lock | $95.0 \pm 4.4$ | $86.7 \pm 4.1$ | $93.3 \pm 6.5$ | $81.7 \pm 9.4$ | $90.0 \pm 5.1$ | $85.0 \pm 6.7$ |
| Door-open | $98.3 \pm 3.3$ | $93.3 \pm 9.7$ | $93.3 \pm 6.5$ | $90.0 \pm 7.2$ | $93.3 \pm 4.1$ | $61.7 \pm 24.5$ |
| Door-unlock | $96.7 \pm 4.1$ | $90.0 \pm 7.2$ | $91.7 \pm 6.0$ | $90.0 \pm 5.1$ | $96.7 \pm 4.1$ | $86.7 \pm 8.3$ |
| Drawer-close | $100.0 \pm 0.0$ | $100.0 \pm 0.0$ | $98.3 \pm 3.3$ | $98.3 \pm 3.3$ | $98.3 \pm 3.3$ | $98.3 \pm 3.3$ |
| Drawer-open | $50.0 \pm 12.4$ | $33.3 \pm 10.9$ | $31.7 \pm 11.8$ | $30.0 \pm 14.3$ | $21.7 \pm 14.7$ | $45.0 \pm 13.1$ |
| Faucet-close | $96.7 \pm 4.1$ | $95.0 \pm 6.7$ | $83.3 \pm 10.9$ | $83.3 \pm 18.0$ | $90.0 \pm 7.2$ | $78.3 \pm 11.8$ |
| Faucet-open | $90.0 \pm 5.1$ | $90.0 \pm 10.1$ | $68.3 \pm 24.0$ | $88.3 \pm 3.3$ | $80.0 \pm 7.2$ | $65.0 \pm 20.7$ |
| Hammer | $33.3 \pm 14.0$ | $20.0 \pm 11.3$ | $33.3 \pm 10.9$ | $36.7 \pm 15.7$ | $40.0 \pm 16.0$ | $40.0 \pm 13.4$ |
| Hand-insert | $30.0 \pm 11.3$ | $16.7 \pm 10.9$ | $26.7 \pm 8.3$ | $25.0 \pm 8.4$ | $15.0 \pm 4.4$ | $25.0 \pm 8.4$ |
| Handle-press | $98.3 \pm 3.3$ | $91.7 \pm 6.0$ | $93.3 \pm 4.1$ | $93.3 \pm 6.5$ | $91.7 \pm 7.9$ | $93.3 \pm 4.1$ |
| Handle-press-side | $95.0 \pm 4.4$ | $88.3 \pm 7.9$ | $85.0 \pm 8.4$ | $93.3 \pm 9.7$ | $93.3 \pm 9.7$ | $88.3 \pm 7.9$ |
| Handle-pull | $\mathbf{68.3} \pm 12.8$ | $45.0 \pm 11.0$ | $26.7 \pm 9.7$ | $40.0 \pm 10.1$ | $36.7 \pm 4.1$ | $50.0 \pm 14.3$ |
| Handle-pull-side | $75.0 \pm 15.8$ | $78.3 \pm 13.8$ | $36.7 \pm 6.5$ | $68.3 \pm 16.3$ | $66.7 \pm 21.3$ | $58.3 \pm 9.4$ |
| Lever-pull | $25.0 \pm 11.0$ | $21.7 \pm 6.0$ | $23.3 \pm 9.7$ | $25.0 \pm 4.4$ | $25.0 \pm 8.4$ | $28.3 \pm 6.0$ |
| Peg-insert-side | $\mathbf{41.7} \pm 10.6$ | $23.3 \pm 10.9$ | $25.0 \pm 13.1$ | $18.3 \pm 6.0$ | $26.7 \pm 9.7$ | $23.3 \pm 8.3$ |
| Peg-unplug-side | $71.7 \pm 12.8$ | $68.3 \pm 10.6$ | $51.7 \pm 22.3$ | $60.0 \pm 11.3$ | $58.3 \pm 15.5$ | $50.0 \pm 13.4$ |
| Pick-out-of-hole | $25.0 \pm 6.7$ | $26.7 \pm 14.0$ | $18.3 \pm 9.4$ | $30.0 \pm 7.2$ | $28.3 \pm 17.1$ | $0.0 \pm 0.0$ |
| Pick-place | $1.7 \pm 2.2$ | $3.3 \pm 2.8$ | $0.0 \pm 0.0$ | $0.0 \pm 0.0$ | $1.7 \pm 2.2$ | $0.0 \pm 0.0$ |
| Pick-place-wall | $0.0 \pm 0.0$ | $5.0 \pm 6.7$ | $1.7 \pm 3.3$ | $1.7 \pm 3.3$ | $3.3 \pm 4.1$ | $0.0 \pm 0.0$ |
| Plate-slide | $56.7 \pm 12.0$ | $56.7 \pm 12.0$ | $48.3 \pm 11.8$ | $60.0 \pm 8.8$ | $53.3 \pm 14.9$ | $63.3 \pm 9.7$ |
| Plate-slide-back | $\mathbf{38.3} \pm 9.4$ | $23.3 \pm 18.0$ | $20.0 \pm 13.4$ | $15.0 \pm 15.0$ | $15.0 \pm 9.8$ | $8.3 \pm 10.6$ |
| Plate-slide-back-side | $74.0 \pm 17.5$ | $66.7 \pm 6.5$ | $74.0 \pm 8.3$ | $68.0 \pm 25.1$ | $63.3 \pm 13.1$ | $76.7 \pm 8.3$ |
| Plate-slide-side | $75.0 \pm 23.3$ | $61.7 \pm 24.0$ | $61.7 \pm 13.8$ | $53.3 \pm 23.6$ | $66.7 \pm 12.0$ | $63.3 \pm 14.0$ |
| Push | $\mathbf{38.3} \pm 11.8$ | $13.3 \pm 4.1$ | $20.0 \pm 7.2$ | $16.7 \pm 6.5$ | $8.3 \pm 12.8$ | $13.3 \pm 9.7$ |
| Push-back | $25.0 \pm 5.7$ | $20.0 \pm 4.8$ | $25.0 \pm 5.7$ | $16.7 \pm 4.4$ | $21.7 \pm 5.3$ | $16.7 \pm 5.6$ |
| Push-wall | $\mathbf{60.0} \pm 11.3$ | $21.7 \pm 7.9$ | $41.7 \pm 10.6$ | $21.7 \pm 10.6$ | $21.7 \pm 7.9$ | $36.7 \pm 15.7$ |
| Reach | $6.7 \pm 3.3$ | $5.0 \pm 4.4$ | $3.3 \pm 4.1$ | $1.7 \pm 3.3$ | $6.7 \pm 9.7$ | $1.7 \pm 3.3$ |
| Reach-wall | $6.7 \pm 4.1$ | $3.3 \pm 4.1$ | $5.0 \pm 4.4$ | $0.0 \pm 0.0$ | $4.0 \pm 4.8$ | $5.0 \pm 6.7$ |
| Shelf-place | $15.0 \pm 11.0$ | $5.0 \pm 6.7$ | $6.7 \pm 6.5$ | $5.0 \pm 6.7$ | $5.0 \pm 6.7$ | $1.7 \pm 3.3$ |
| Soccer | $38.3 \pm 6.0$ | $18.3 \pm 7.9$ | $21.7 \pm 9.4$ | $20.0 \pm 10.1$ | $10.0 \pm 0.0$ | $30.0 \pm 13.4$ |
| Stick-pull | $6.7 \pm 8.3$ | $0.0 \pm 0.0$ | $0.0 \pm 0.0$ | $0.0 \pm 0.0$ | $0.0 \pm 0.0$ | $1.7 \pm 3.3$ |
| Stick-push | $0.0 \pm 0.0$ | $5.0 \pm 4.4$ | $0.0 \pm 0.0$ | $1.7 \pm 3.3$ | $1.7 \pm 3.3$ | $5.0 \pm 6.7$ |
| Sweep | $88.3 \pm 8.3$ | $75.0 \pm 8.4$ | $91.7 \pm 6.0$ | $76.7 \pm 9.7$ | $56.7 \pm 8.3$ | $88.3 \pm 9.4$ |
| Sweep-into | $88.3 \pm 9.4$ | $68.3 \pm 14.7$ | $65.0 \pm 15.0$ | $58.3 \pm 15.5$ | $60.0 \pm 18.2$ | $85.0 \pm 13.1$ |
| Window-close | $100.0 \pm 0.0$ | $95.0 \pm 4.4$ | $95.0 \pm 6.7$ | $98.3 \pm 3.3$ | $100.0 \pm 0.0$ | $98.3 \pm 3.3$ |
| Window-open | $\mathbf{100.0} \pm 0.0$ | $88.3 \pm 6.0$ | $78.3 \pm 11.8$ | $81.7 \pm 6.0$ | $73.3 \pm 9.7$ | $81.7 \pm 10.6$ |

Table 8 summarize zero-shot adaptation performance for in-distribution tasks and Table 9 summarize zero-shot adaptation performance for out-of-distribution tasks. For environments where some varia-

Table 8: Zero-shot in-distribution performance on MuJoCo and Contextual-DMC benchmarks. Average returns over 6 random seeds, $\pm$ represents $95\%$ confidence intervals. **Bold** indicates the highest value with statistical significance according to the t-test with p-value $< 0.05$.

| Environment | C-DCWM | CSRO | DORA | FOCAL | UNICORN-SS | UNICORN-SUP |
|---|---|---|---|---|---|---|
| Ant-dir | $726.7 \pm 38.1$ | $699.3 \pm 26.6$ | $526.9 \pm 28.0$ | $678.4 \pm 40.3$ | $668.1 \pm 46.4$ | $366.6 \pm 41.6$ |
| Cheetah-LS | $\mathbf{935.0} \pm 11.9$ | $828.3 \pm 34.9$ | $901.7 \pm 25.3$ | $825.3 \pm 36.6$ | $794.5 \pm 44.5$ | $841.9 \pm 50.9$ |
| Cheetah-speed | $\mathbf{706.4} \pm 33.1$ | $556.5 \pm 31.3$ | $497.1 \pm 44.0$ | $447.3 \pm 73.0$ | $490.2 \pm 82.1$ | $447.9 \pm 49.1$ |
| Finger-LS | $\mathbf{972.0} \pm 5.0$ | $897.8 \pm 41.8$ | $869.2 \pm 46.4$ | $863.0 \pm 58.3$ | $885.3 \pm 49.6$ | $824.0 \pm 30.6$ |
| Finger-speed | $\mathbf{943.3} \pm 8.4$ | $773.7 \pm 48.5$ | $492.4 \pm 51.7$ | $746.7 \pm 49.3$ | $671.9 \pm 54.6$ | $614.4 \pm 45.0$ |
| Hopper-mass | $566.0 \pm 13.5$ | $450.8 \pm 79.2$ | $555.0 \pm 20.6$ | $535.0 \pm 33.3$ | $533.7 \pm 38.3$ | $491.6 \pm 145.6$ |
| Walker-friction | $\mathbf{578.2} \pm 13.6$ | $503.7 \pm 39.5$ | $513.9 \pm 29.1$ | $522.0 \pm 32.8$ | $522.1 \pm 21.6$ | $476.7 \pm 32.4$ |
| Walker-LS | $937.2 \pm 9.9$ | $882.5 \pm 100.3$ | $885.5 \pm 40.8$ | $898.9 \pm 30.3$ | $900.0 \pm 49.2$ | $889.9 \pm 36.9$ |
| Walker-speed | $829.7 \pm 53.5$ | $767.1 \pm 31.7$ | $446.3 \pm 43.8$ | $653.1 \pm 99.4$ | $598.6 \pm 54.4$ | $513.1 \pm 48.7$ |

Table 9: Zero-shot out-of-distribution performance on MuJoCo and Contextual-DMC benchmarks. Average returns over 6 random seeds, $\pm$ represents $95\%$ confidence intervals. **Bold** indicates the highest value with statistical significance according to the t-test with p-value $< 0.05$.

| Environment | C-DCWM | CSRO | DORA | FOCAL | UNICORN-SS | UNICORN-SUP |
|---|---|---|---|---|---|---|
| Ant-dir | $410.7 \pm 36.9$ | $399.2 \pm 63.9$ | $156.8 \pm 44.7$ | $368.8 \pm 64.2$ | $405.9 \pm 38.5$ | $-211.4 \pm 195.5$ |
| Cheetah-LS | $\mathbf{865.5} \pm 20.4$ | $813.9 \pm 28.1$ | $785.8 \pm 39.6$ | $826.6 \pm 20.6$ | $806.1 \pm 28.6$ | $795.8 \pm 44.3$ |
| Cheetah-speed | $\mathbf{756.0} \pm 92.5$ | $603.5 \pm 96.5$ | $573.0 \pm 59.0$ | $607.8 \pm 160.4$ | $598.8 \pm 103.4$ | $554.8 \pm 75.9$ |
| Finger-LS | $\mathbf{886.7} \pm 11.8$ | $762.7 \pm 57.9$ | $717.8 \pm 53.3$ | $786.8 \pm 32.6$ | $816.1 \pm 47.7$ | $691.5 \pm 54.8$ |
| Finger-speed | $\mathbf{948.1} \pm 9.3$ | $822.8 \pm 32.3$ | $532.9 \pm 85.5$ | $771.3 \pm 39.0$ | $709.6 \pm 43.6$ | $675.3 \pm 66.0$ |
| Hopper-mass | $\mathbf{583.4} \pm 4.1$ | $543.6 \pm 32.1$ | $534.2 \pm 23.3$ | $547.3 \pm 13.3$ | $550.7 \pm 11.3$ | $463.9 \pm 133.6$ |
| Walker-friction | $474.1 \pm 30.3$ | $475.1 \pm 22.5$ | $462.4 \pm 26.5$ | $473.7 \pm 36.5$ | $484.6 \pm 25.0$ | $435.2 \pm 53.8$ |
| Walker-LS | $\mathbf{788.0} \pm 27.8$ | $611.5 \pm 31.6$ | $650.9 \pm 42.5$ | $658.0 \pm 41.0$ | $657.6 \pm 41.7$ | $649.9 \pm 50.7$ |
| Walker-speed | $\mathbf{831.5} \pm 44.5$ | $767.2 \pm 24.0$ | $425.3 \pm 56.1$ | $659.6 \pm 120.1$ | $623.7 \pm 98.0$ | $535.5 \pm 44.2$ |

tion factors in the out-of-distribution tasks interpolate those seen during training (*e.g.*, [Cheetah, Finger, Walker]-speed), the performance on out-of-distribution tasks is relatively close to in-distribution performance. In contrast, when out-of-distribution generalization requires extrapolating beyond the training variation factors (*e.g.*, [Cheetah, Finger, Walker]-LS, Ant-Dir), a larger performance gap between in-distribution and out-of-distribution tasks is observed. C-DCWM performs more consistently compared to baselines when generalizing to out-of-distribution tasks. These results indicate that latent temporal consistency can improve performance on in-distribution tasks by converting the observation space to a latent space and can increase generalization to out-of-distribution tasks by encouraging the context encoder to capture latent dynamics.

## B.2 DISENTANGLEMENT METRICS

Table 10 and Table 11 illustrate disentanglement metrics for Finger-LS and Walker-LS environments respectively. We observe similar trends as Table 3, where training the context encoder based on world modeling (WM) results in higher disentanglement than training the context encoder with reconstruction (UNICORN-SUP), and including contrastive learning can improve task distinguishability (informativeness, explicitness). However, while the reconstruction objective (UNICORN-SUP) results in higher disentanglement than contrastive objectives (FOCAL, InfoNCE) in Cheetah-LS (Table 3), in [Finger, Walker]-LS reconstruction objective results in a lower disentanglement score.

## B.3 ABLATION: CONTRASTIVE LEARNING

In this section, we perform an ablation study on the contrastive and world modeling objectives used to train the context encoder. We compare two contrastive learning objectives commonly employed in OMRL: InfoNCE and FOCAL Li et al. (2020) (also referred to as distance metric learning). The FOCAL objective is defined as:

$$\mathcal{L}_{\text{FOCAL}}(\phi) = \mathbb{1}\{i = j\}\|z^i - z^j\|_2^2 + \mathbb{1}\{i \neq j\}\frac{\beta}{\|z^i - z^j\|_2^2 + \epsilon_0}. \tag{14}$$

For a fair comparison, we convert the observation space to a discrete latent space for all the methods with the same world model (DCWM). Table 12 reports few-shot in-distribution testing and Table 13

Table 10: Disentanglement metrics (DCI, InfoMEC) for the Finger Length/Speed environment. Average metrics over 6 random seeds, $\pm$ represents 95% confidence intervals.

| | Disentanglement | Completeness | Informativeness | Modularity | Explicitness | Compactness |
|---|---|---|---|---|---|---|
| FOCAL | $0.36 \pm 0.06$ | $0.30 \pm 0.07$ | $0.70 \pm 0.01$ | $0.83 \pm 0.02$ | $0.80 \pm 0.01$ | $0.24 \pm 0.01$ |
| CSRO | $0.36 \pm 0.03$ | $0.38 \pm 0.08$ | $0.71 \pm 0.02$ | $0.82 \pm 0.02$ | $0.79 \pm 0.01$ | $0.24 \pm 0.01$ |
| DORA (InfoNCE) | $0.41 \pm 0.09$ | $0.43 \pm 0.09$ | $0.64 \pm 0.03$ | $0.77 \pm 0.03$ | $0.79 \pm 0.02$ | $0.23 \pm 0.01$ |
| UNICORN-SS | $0.33 \pm 0.05$ | $0.31 \pm 0.08$ | $0.69 \pm 0.02$ | $0.79 \pm 0.02$ | $0.78 \pm 0.01$ | $0.15 \pm 0.01$ |
| UNICORN-SUP | $0.25 \pm 0.05$ | $0.34 \pm 0.06$ | $0.57 \pm 0.01$ | $0.85 \pm 0.02$ | $0.73 \pm 0.01$ | $0.30 \pm 0.04$ |
| WM | $0.41 \pm 0.03$ | $0.43 \pm 0.02$ | $0.70 \pm 0.01$ | $0.82 \pm 0.02$ | $0.85 \pm 0.01$ | $0.23 \pm 0.01$ |
| WM+FOCAL | $0.41 \pm 0.06$ | $0.43 \pm 0.04$ | $0.80 \pm 0.04$ | $0.83 \pm 0.02$ | $0.86 \pm 0.02$ | $0.23 \pm 0.01$ |
| WM+InfoNCE | $0.46 \pm 0.06$ | $0.43 \pm 0.04$ | $0.82 \pm 0.03$ | $0.82 \pm 0.02$ | $0.87 \pm 0.02$ | $0.23 \pm 0.01$ |

Table 11: Disentanglement metrics (DCI, InfoMEC) for the Walker Length/Speed environment. Average metrics over 6 random seeds, $\pm$ represents 95% confidence intervals.

| | Disentanglement | Completeness | Informativeness | Modularity | Explicitness | Compactness |
|---|---|---|---|---|---|---|
| FOCAL | $0.33 \pm 0.07$ | $0.31 \pm 0.06$ | $0.83 \pm 0.04$ | $0.69 \pm 0.03$ | $0.82 \pm 0.02$ | $0.23 \pm 0.01$ |
| CSRO | $0.42 \pm 0.08$ | $0.41 \pm 0.09$ | $0.79 \pm 0.03$ | $0.72 \pm 0.03$ | $0.82 \pm 0.02$ | $0.23 \pm 0.01$ |
| DORA (InfoNCE) | $0.22 \pm 0.07$ | $0.17 \pm 0.14$ | $0.67 \pm 0.05$ | $0.68 \pm 0.05$ | $0.76 \pm 0.03$ | $0.23 \pm 0.01$ |
| UNICORN-SS | $0.38 \pm 0.07$ | $0.31 \pm 0.05$ | $0.84 \pm 0.03$ | $0.73 \pm 0.03$ | $0.84 \pm 0.01$ | $0.24 \pm 0.01$ |
| UNICORN-SUP | $0.20 \pm 0.03$ | $0.18 \pm 0.07$ | $0.38 \pm 0.04$ | $0.80 \pm 0.04$ | $0.64 \pm 0.02$ | $0.26 \pm 0.02$ |
| WM | $0.39 \pm 0.05$ | $0.27 \pm 0.06$ | $0.75 \pm 0.04$ | $0.79 \pm 0.05$ | $0.78 \pm 0.03$ | $0.24 \pm 0.02$ |
| WM+FOCAL | $0.43 \pm 0.04$ | $0.29 \pm 0.02$ | $0.88 \pm 0.05$ | $0.77 \pm 0.05$ | $0.86 \pm 0.03$ | $0.23 \pm 0.01$ |
| WM+InfoNCE | $0.44 \pm 0.09$ | $0.30 \pm 0.09$ | $0.84 \pm 0.04$ | $0.79 \pm 0.04$ | $0.88 \pm 0.03$ | $0.24 \pm 0.01$ |

reports few-shot out-of-distribution testing. Here, *WM* refers to training the context encoder with the world modeling objective, and + indicates the combination of objectives. For environments in Meta-World benchmarks (last 6 rows in Table 12), we observe no significant difference in the performance of different objectives. Training the context encoder solely with the world modeling objective (WM) is insufficient, as it fails to distinguish between different tasks. This limitation is particularly pronounced in environments where variation factors affect only the reward function, rather than the transition dynamics (*e.g.*, Ant-dir, where the desired forward direction varies, or environments requiring the inference of desired speed). Only using contrastive learning results in good performance across most tasks, while InfoNCE outperforms FOCAL significantly in certain environments, especially on out-of-distribution testing. Adding the world modeling objective to the contrastive objective has an insignificant impact on in-distribution performance; however, it can improve generalization to out-of-distribution tasks for certain environments. We employ the same relative weighting of the contrastive objective with respect to the world modeling objective across all environments. Overall, combining InfoNCE with the world modeling objective produces more robust results across environments compared to combining FOCAL with the world modeling objective.

Table 12: Ablation on contrastive learning and world modeling, few-shot in-distribution performance. Average returns/success rates over 6 random seeds, $\pm$ represents 95% confidence intervals. **Bold** indicates the highest value with statistical significance according to the t-test with p-value $< 0.05$.

| Environment | FOCAL | InfoNCE | WM | WM+FOCAL | WM+InfoNCE |
|---|---|---|---|---|---|
| Ant-dir | $841.6 \pm 31.1$ | $857.7 \pm 42.3$ | $487.5 \pm 91.8$ | $859.0 \pm 20.5$ | $863.1 \pm 36.2$ |
| Cheetah-LS | $940.0 \pm 16.2$ | $937.7 \pm 17.7$ | $941.5 \pm 16.6$ | $933.2 \pm 15.0$ | $944.8 \pm 4.9$ |
| Cheetah-speed | $721.1 \pm 54.2$ | $727.3 \pm 23.8$ | $395.0 \pm 36.2$ | $711.2 \pm 94.0$ | $764.1 \pm 39.2$ |
| Finger-LS | $971.0 \pm 10.5$ | $968.4 \pm 10.8$ | $974.6 \pm 5.5$ | $973.7 \pm 2.8$ | $968.0 \pm 5.5$ |
| Finger-speed | $789.7 \pm 189.6$ | $958.1 \pm 5.7$ | $706.1 \pm 150.3$ | $770.0 \pm 185.2$ | $\mathbf{967}.4 \pm 2.0$ |
| Walker-LS | $929.8 \pm 24.4$ | $947.7 \pm 15.1$ | $904.5 \pm 24.9$ | $928.9 \pm 21.6$ | $934.6 \pm 20.1$ |
| Walker-speed | $622.6 \pm 65.9$ | $842.2 \pm 35.6$ | $522.1 \pm 95.4$ | $552.0 \pm 82.4$ | $835.7 \pm 37.3$ |
| Button-press | $100.0 \pm 0.0$ | $100.0 \pm 0.0$ | $98.3 \pm 3.3$ | $100.0 \pm 0.0$ | $100.0 \pm 0.0$ |
| Coffee-button | $100.0 \pm 0.0$ | $100.0 \pm 0.0$ | $100.0 \pm 0.0$ | $100.0 \pm 0.0$ | $100.0 \pm 0.0$ |
| Dial-turn | $93.3 \pm 4.1$ | $93.3 \pm 6.5$ | $98.3 \pm 3.3$ | $96.7 \pm 4.1$ | $98.3 \pm 3.3$ |
| Door-open | $100.0 \pm 0.0$ | $100.0 \pm 0.0$ | $100.0 \pm 0.0$ | $96.7 \pm 6.5$ | $100.0 \pm 0.0$ |
| Door-unlock | $98.3 \pm 3.3$ | $95.0 \pm 6.7$ | $98.3 \pm 3.3$ | $98.3 \pm 3.3$ | $100.0 \pm 0.0$ |
| Handle-press | $96.7 \pm 4.1$ | $96.7 \pm 4.1$ | $95.0 \pm 6.7$ | $95.0 \pm 6.7$ | $98.3 \pm 3.3$ |

Table 13: Ablation on contrastive learning and world modeling, few-shot out-of-distribution performance. Average returns over 6 random seeds, $\pm$ represents 95% confidence intervals. **Bold** indicates the highest value with statistical significance according to the t-test with p-value $< 0.05$. **Combining contrastive learning with latent temporal consistency enhances generalization to out-of-distribution tasks**.

| Environment | FOCAL | InfoNCE | WM | WM+FOCAL | WM+InfoNCE |
|---|---|---|---|---|---|
| Ant-dir | $529.0 \pm 41.0$ | $363.2 \pm 173.9$ | $203.6 \pm 102.9$ | $540.8 \pm 111.2$ | $401.8 \pm 92.4$ |
| Cheetah-LS | $864.4 \pm 21.7$ | $864.7 \pm 29.9$ | $867.2 \pm 7.5$ | $876.1 \pm 24.7$ | $860.6 \pm 10.2$ |
| Cheetah-speed | $729.0 \pm 42.4$ | $754.9 \pm 70.6$ | $486.8 \pm 111.1$ | $908.7 \pm 74.2$ | $\mathbf{967.7} \pm 10.5$ |
| Finger-LS | $836.3 \pm 45.5$ | $809.6 \pm 53.1$ | $838.3 \pm 64.8$ | $860.0 \pm 42.7$ | $850.9 \pm 41.5$ |
| Finger-speed | $793.8 \pm 189.0$ | $868.5 \pm 10.3$ | $766.6 \pm 176.2$ | $755.7 \pm 185.0$ | $\mathbf{978.5} \pm 6.0$ |
| Walker-LS | $740.2 \pm 52.9$ | $793.2 \pm 62.2$ | $757.0 \pm 65.2$ | $738.9 \pm 51.8$ | $792.3 \pm 41.3$ |
| Walker-speed | $619.4 \pm 111.4$ | $782.7 \pm 32.2$ | $507.7 \pm 103.4$ | $568.7 \pm 128.5$ | $\mathbf{833.2} \pm 64.5$ |

## B.4 ABLATION: NUMBER OF PARAMETERS

In this section, we investigate how increasing the number of trainable parameters affects performance across different methods. C-DCWM maps the observation space to a latent space using a latent world model, which increases the total number of parameters. Fig. 6 illustrates the few-shot in-distribution performance of each method for different model sizes. For the baselines, we vary the number of hidden layers in $\{2, 3\}$ and the number of hidden units in $\{256, 512, 1024\}$, resulting in six model sizes (default is two hidden layers with 256 neurons). To ensure architectural consistency with C-DCWM, we apply Layer Normalization and the Mish activation function across all baseline networks. For C-DCWM, we set the latent dimension and the number of neurons in the dynamic head of the latent world model to $\{64, 128, 256, 512\}$ while setting the number of neurons in the offline RL (IQL) networks to $\{64, 128, 256, 512\}$ (6 combinations, default is 512 latent dimensions and neurons in the dynamic head, 256 neurons for the offline RL networks). C-DCWM exhibits better scaling with model size: performance generally improves as the number of parameters increases. However, for smaller model sizes, C-DCWM underperforms the baselines in some environments.

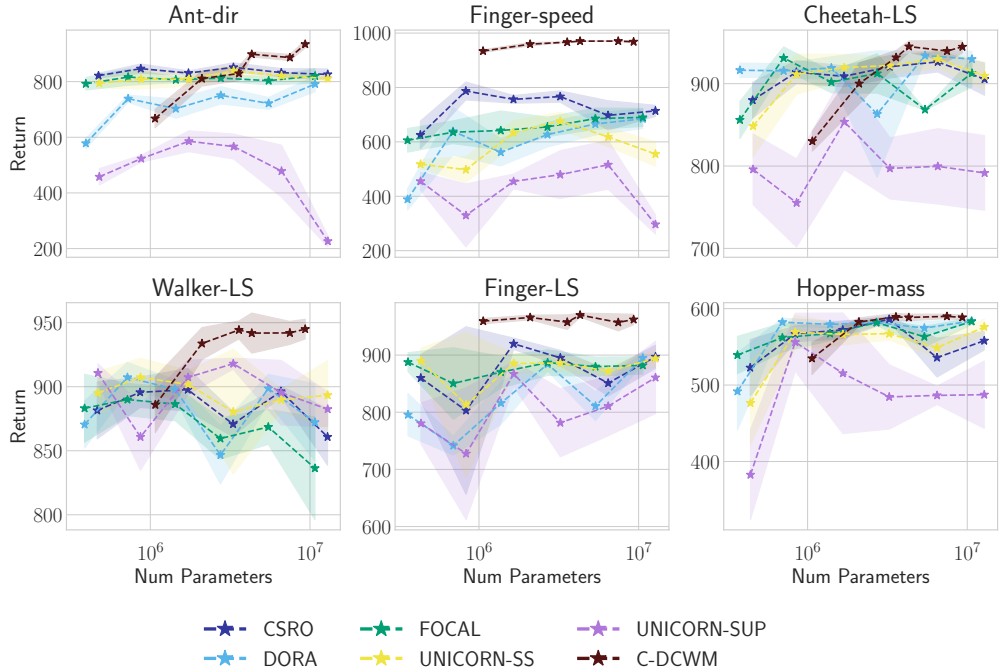

Figure 6: **C-DCWM scales more effectively.** Few-shot generalization to in-distribution tasks for different model sizes. The shaded area represents the standard deviation over 6 random seeds.

Table 14: **Ablation: different offline RL methods for policy optimizations.** Few-shot generalization to in-distribution tasks. Average returns/success rates over 6 random seeds, $\pm$ represents 95% confidence intervals. **Bold** indicates the highest value with statistical significance according to the t-test with p-value $< 0.05$.

| Environment | CQL | IQL (Def) | TD3+BC |
|---|---|---|---|
| Ant-dir | $780.7 \pm 54.4$ | $\mathbf{863.1} \pm 36.2$ | $731.8 \pm 29.8$ |
| Cheetah-LS | $949.8 \pm 12.1$ | $943.0 \pm 11.5$ | $846.2 \pm 60.6$ |
| Cheetah-speed | $\mathbf{813.3} \pm 3.5$ | $751.2 \pm 27.9$ | $620.8 \pm 133.0$ |
| Finger-LS | $\mathbf{987.5} \pm 3.8$ | $957.1 \pm 23.9$ | $330.7 \pm 26.7$ |
| Finger-speed | $970.2 \pm 4.3$ | $962.0 \pm 9.2$ | $208.7 \pm 31.8$ |
| Hopper-mass | $584.2 \pm 1.1$ | $587.5 \pm 4.9$ | $159.4 \pm 26.1$ |
| Walker-friction | $585.6 \pm 32.6$ | $563.6 \pm 33.5$ | $535.6 \pm 26.6$ |
| Walker-LS | $948.0 \pm 7.2$ | $937.1 \pm 16.6$ | $710.2 \pm 84.9$ |
| Walker-speed | $814.0 \pm 42.4$ | $827.6 \pm 34.6$ | $655.7 \pm 108.2$ |
| Button-press | $95.0 \pm 9.8$ | $100.0 \pm 0.0$ | $80.0 \pm 5.1$ |
| Coffee-button | $100.0 \pm 0.0$ | $100.0 \pm 0.0$ | $91.7 \pm 7.9$ |
| Dial-turn | $90.0 \pm 19.6$ | $91.7 \pm 10.6$ | $95.0 \pm 4.4$ |
| Door-open | $100.0 \pm 0.0$ | $100.0 \pm 0.0$ | $23.3 \pm 4.1$ |
| Door-unlock | $100.0 \pm 0.0$ | $100.0 \pm 0.0$ | $51.7 \pm 10.6$ |
| Handle-press | $60.0 \pm 19.6$ | $\mathbf{93.3} \pm 4.1$ | $76.7 \pm 6.5$ |

## B.5 ABLATION: OFFLINE RL

As described in Sec. 3.2, policy optimization with offline data requires regularization to avoid OOD action selection when computing the target for the value function. Offline RL methods address this issue in different ways, and in principle, any offline RL method can be used for policy optimization in C-DCWM. By default, we use Implicit Q-Learning (IQL) for all methods, which predicts an upper expectile of the TD targets in SARSA style without querying OOD actions. We also evaluate C-DCWM with Conservative Q-Learning (CQL, Kumar et al. 2020) and TD3+BC (Fujimoto & Gu, 2021) for policy optimization, summarized in Table 14. CQL regularizes the value function by reducing the q-value for OOD actions, resulting in a pessimistic value function. TD3+BC, on the other hand, regularizes the policy to stay close to the behavior policy by adding a behavior cloning objective to the policy optimization. We used one set of hyperparameters (default values) for all methods without further fine-tuning. We find that IQL in general is more robust, performing well in diverse environments. CQL generally performs on par with IQL, even outperforming significantly in two environments. However, the computation cost of CQL is generally higher than IQL. We sometimes observe a performance drop when training for a larger number of steps. TD3+BC generally has a lower performance than CQL and IQL in our settings. We hypothesize that fine-tuning the regularization weight for each environment can increase the performance.

## B.6 COMPUTATION COST

Fig. 7 compares the computation cost for different methods. All experiments are conducted using the same hardware, as described in Sec. A, to ensure a fair comparison. Although C-DCWM has a longer training time per step, it generally converges faster than the baselines, compensating for the higher per-step computational cost. During testing, C-DCWM is slightly slower because it first maps the observation to the latent space using the observation encoder, after which the policy produces actions.

UNICORN-SUP trains the context encoder solely using the prediction loss and has the lowest computational cost per training step. However, incorporating contrastive learning can improve task representation learning and, consequently, generalization to new tasks. DORA, which uses the InfoNCE loss for contrastive learning, has a lower per-step computational cost than FOCAL, which uses distance metric learning, suggesting that InfoNCE is more computationally efficient than the FOCAL objectives in Eq. (14). CSRO and UNICORN-SS aim to reduce context distribution shift by minimizing a CLUB upper bound of mutual information and by adding a prediction loss, respectively. These approaches require additional networks, increasing their computational cost per training step. During test time, all baselines have the same computational cost since the policy and context encoder architectures are identical across the baseline methods.

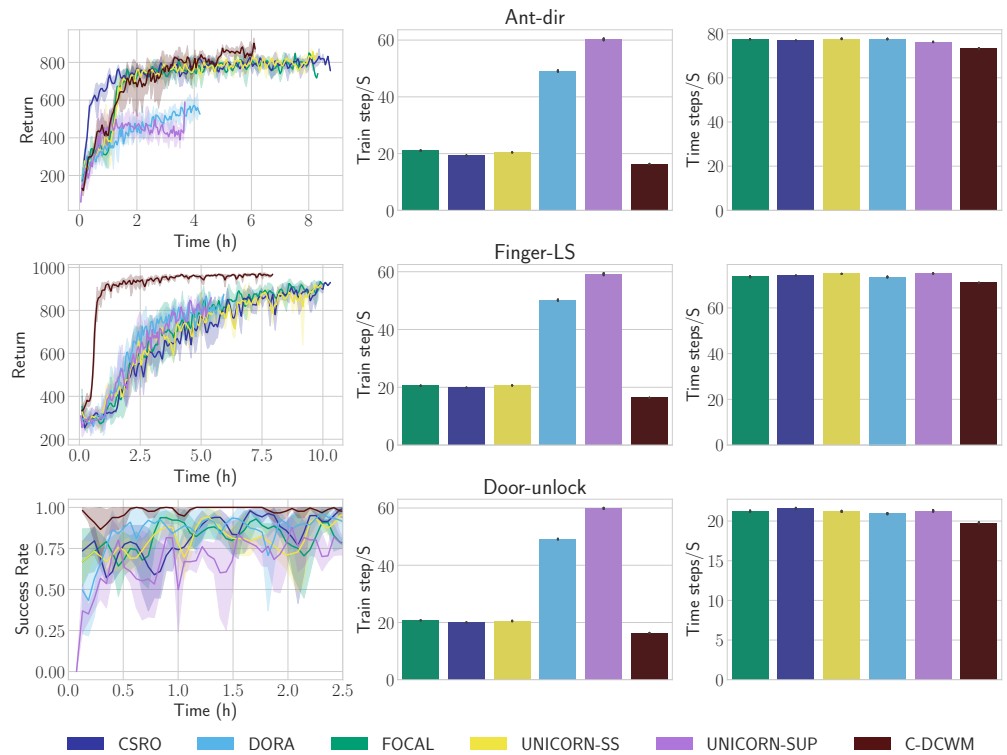

Figure 7: Comparing the computation cost. **Left:** Few-shot performance on in-distribution tasks with training time. **Middle:** Number of training steps (full backpropagation and updating the networks) per second. **Right:** Number of time steps per second during testing. **C-DCWM is computationally more expensive during training, yet it converges faster.** Mapping the observation space to the latent space adds insignificant computation overhead during testing. Results are averaged over 3 random seeds while considering the standard deviation.

## B.7    COMPARISON TO DREAMERV3

DreamerV3 (Hafner et al., 2025) is a model-based RL method that employs a Recurrent State-Space Model (RSSM; Doerr et al., 2018) for latent dynamics while jointly predicting rewards, observations, and terminations. Its latent space includes both continuous and discrete variables, resembling the discrete latent space used in C-DCWM. The policy and value function are optimized within the world model. However, DreamerV3 uses one-hot encoding for its discrete latent variables, whereas C-DCWM employs a codebook-based representation. To compare DreamerV3 with C-DCWM in the OMRL setting, we use a public PyTorch implementation[1] with default hyperparameters.

Table 15 reports the zero-shot performance on in-distribution tasks; for DreamerV3, we reset the RSSM hidden state at the initial timestep during meta-testing. DreamerV3 struggles to generalize to new tasks in OMRL settings, particularly in environments where optimal policies differ significantly across tasks. For example, in [Cheetah, Finger, Walker]-speed environments, the agent must move both forward and backward at different speeds, and in Ant-dir environments, the agent must move in different directions. On the other hand, DreamerV3 shows better generalization in environments where optimal task-specific policies are more similar, such as Hopper-mass and Walker-friction.

We also hypothesize that the policy may exploit inaccuracies in the world model, since the world model is trained solely on static datasets. The policy is optimized to maximize expected return under the world model's predictions, without any penalty for acting in uncertain or poorly modeled regions. In online RL settings, the policy's actions would be executed in the real environment, and

---

[1]https://github.com/NM512/dreamerv3-torch

the world model would be updated accordingly; however, this corrective mechanism is absent in the offline OMRL scenario.

Table 15: **DreamerV3 fails to generalize in OMRL settings**. Zero-shot generalization to in-distribution tasks. Average returns/success rates over 3 random seeds, $\pm$ represents $95\%$ confidence intervals.

| Environment | C-DCWM | DreamerV3 |
|---|---|---|
| Ant-dir | $649.9 \pm 50.7$ | $-3.6 \pm 3.0$ |
| Cheetah-LS | $936.5 \pm 10.8$ | $584.8 \pm 53.2$ |
| Cheetah-speed | $664.4 \pm 51.7$ | $178.7 \pm 20.8$ |
| Finger-LS | $966.4 \pm 5.5$ | $438.6 \pm 85.3$ |
| Finger-speed | $946.9 \pm 9.6$ | $187.0 \pm 72.7$ |
| Hopper-mass | $579.9 \pm 9.5$ | $555.1 \pm 3.6$ |
| Walker-friction | $580.6 \pm 4.7$ | $523.6 \pm 41.1$ |
| Walker-LS | $939.7 \pm 8.3$ | $643.6 \pm 46.0$ |
| Walker-speed | $705.8 \pm 70.9$ | $149.9 \pm 10.3$ |
| Button-press | $96.7 \pm 4.1$ | $2.2 \pm 3.2$ |
| Coffee-button | $100.0 \pm 0.0$ | $72.8 \pm 21.1$ |
| Dial-turn | $91.7 \pm 6.0$ | $0.6 \pm 1.1$ |
| Door-open | $100.0 \pm 0.0$ | $0.0 \pm 0.0$ |
| Door-unlock | $100.0 \pm 0.0$ | $0.0 \pm 0.0$ |
| Handle-press | $95.0 \pm 4.4$ | $4.4 \pm 4.0$ |

## B.8 DECISION TIME PLANNING

Planning with the latent world model can improve sample-efficiency in RL (Hansen et al., 2022; 2024; Scannell et al., 2025). We investigate whether decision time planning with our latent world model can outperform policy optimization by changing the observation space in the OMRL setting. A key challenge for model-based RL methods in offline settings is limited dataset coverage, which can lead to inaccurate predictions in certain regions of the state-action space. By discretizing the latent space into fixed codebooks and predicting the next latent state via classification, this issue may be mitigated. Fig. 8 presents the results of decision-time planning across different planning horizons. We use Model Predictive Path Integral (MPPI) for planning, similar to (Hansen et al., 2024; Scannell et al., 2025). For planning, we use the Model Predictive Path Integral (MPPI) method, following Hansen et al. (2024); Scannell et al. (2025). As the planning horizon increases, performance improves, although testing time grows approximately linearly.

We also experimented with incorporating value functions into planning. However, including estimated values trained with IQL for the final step led to a decrease in performance. We hypothesize that value estimates for unobserved state-action pairs are unreliable due to limited dataset coverage. Additionally, IQL does not penalize the Q function for out-of-distribution (OOD) actions; it simply avoids querying them during policy optimization. Planning with value estimates trained using pessimistic methods, such as conservative Q-learning (CQL Kumar et al., 2020), is an interesting future endeavor.

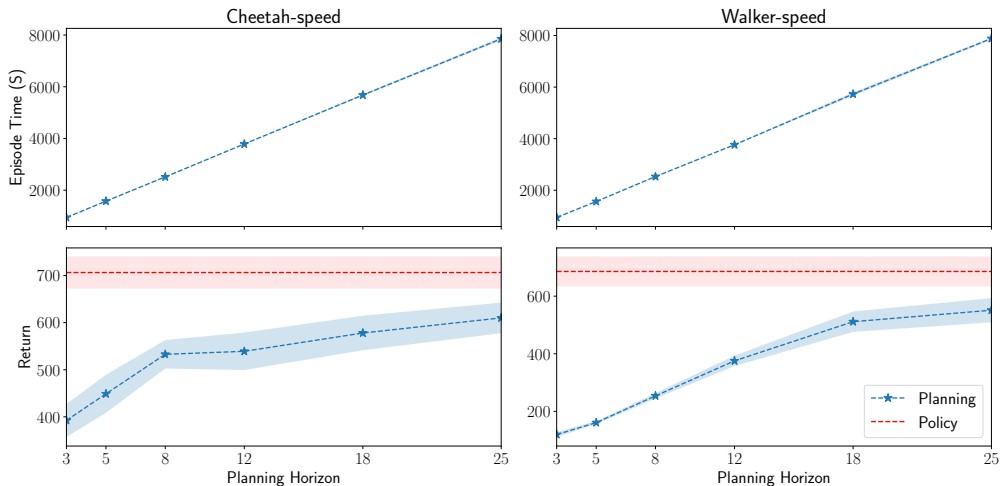

Figure 8: Decision-time planning with our latent world model, zero-shot performance of planning with MPPI. **Policy optimized on the latent space outperforms planning**. A longer planning horizon increases the performance, but planning time also scales linearly, making it unsuitable for real-time control.

