# OpenReview forum: "Contextual Latent World Models for Offline Meta Reinforcement Learning"
_ICLR.cc/2026/Conference — Submitted to ICLR 2026_

### Official Review · Reviewer_iC8p · 2025-10-19

**Soundness:** 3
**Presentation:** 4
**Contribution:** 2
**Rating:** 2
**Confidence:** 4

**Summary:**

This paper introduces C-DCWM, an offline model-based meta-reinforcement learning algorithm. The method utilizes a context-based approach, training a task encoder with an InfoNCE loss. It incorporates the learned task representations into the Discrete Codebook World Model (DCWM), extending it to the multi-task setting. The paper demonstrates that C-DCWM achieves superior performance compared to existing offline meta-rl baselines on the MuJoCo, Contextual-DMC, and Meta-World benchmarks.

**Strengths:**

1. The study of world models capable of generalizing across multiple tasks is of significant practical importance, as it promises to substantially advance the real-world application of reinforcement learning.
2. The authors provide extensive experiments across multiple benchmarks (MuJoCo, Contextual-DMC, and Meta-World) , demonstrating the method's strong empirical performance against several baselines.

**Weaknesses:**

1. The proposed method appears to be a relatively straightforward extension of the DCWM frameworkto the multi-task setting by conditioning it on a task representation. This potentially limits the technical novelty of this work.
2. The evaluation of generalization is limited. The out-of-distribution experiments primarily focus on parametric variations within the same task families. A more challenging and practically relevant OOD evaluation, such as generalization to unseen tasks in Meta-World, is notably absent.
3. The paper lacks a comparative analysis of the computational complexity or overhead (e.g., parameter count, training time) relative to the baseline methods.

**Questions:**

1. The current experiments (e.g., in MuJoC) primarily demonstrate OOD generalization to **new parameters of the same task family**. Does the model also exhibit **cross-task** generalization? For example, if the model is trained on a diverse set of tasks from Meta-World (e.g., 'door-open', 'window-close' et al. without ''window-open'), could it generalize (few-shot or zero-shot) to a completely **unseen** task, such as 'window-open'?
2. Model-based methods typically have a larger parameter count compared to model-free method. Can authors provide analysis (e.g., an ablation study or parameter comparison) to confirm that the superior performance of C-DCWM stems from the effectiveness of the contextual world model itself, and not merely from an increased number of trainable parameters compared to the baselines?
- I will raise my score if my concerns above are addressed.

---

> ### Author Response · Authors · 2025-11-20
> **Response to Reviewer iC8p**
>
> **Q1: The current experiments (e.g., in MuJoCo) primarily demonstrate OOD generalization to new parameters of the same task family. Does the model also exhibit cross-task generalization?**
>
> We thank the reviewer for raising this important question about more challenging forms of out-of-distribution generalization. In response, we expanded our discussion and highlighted results on Meta-World ML10 and ML45 settings, where cross-environment generalization is explicitly evaluated. We report these results in Table 2 (Section 4.1).
>
> **Findings:**
> - C-DCWM outperforms all baselines on both training and held-out testing environments in ML10 and ML45 settings.
> - Performance on testing environments is slightly higher in ML45, consistent with the intuition that a more diverse training set encourages broader generalization.
> - However, we observe a significant performance drop when generalizing to entirely new environments, especially compared to generalization within a single environment family.
>
> We thank the reviewer for prompting us to emphasize this analysis. It clarified the current limits of cross-environment generalization and helped contextualize our method’s strengths and remaining challenges.
>
> **Q2: Model-based methods typically have a larger parameter count compared to model-free methods. Can authors provide analysis (e.g., an ablation study or parameter comparison) to confirm that the superior performance of C-DCWM stems from the effectiveness of the contextual world model itself, and not merely from an increased number of trainable parameters compared to the baselines?**
>
> We appreciate the reviewer’s concern and agree that parameter count could confound performance comparisons. Motivated by this feedback, we conducted additional analysis using six different model sizes, with results shown in Figure 6 (Section B.4).
>
> **Findings:**
> - C-DCWM shows more consistent scaling behavior: performance generally improves as model size increases.
> - In contrast, baselines do not consistently benefit from increased parameter counts.
> - C-DCWM, with the default model size, outperforms baselines across all model sizes.
> - At smaller model sizes, C-DCWM sometimes underperforms baselines with a similar parameter count, as we use smaller policy and value networks to accommodate the discrete world model.
>
> This analysis, added in direct response to the reviewer’s suggestion, significantly strengthened the argument that the contextual world model is responsible for the observed gains.
>
> **Q3: The paper lacks a comparative analysis of the computational complexity or overhead (e.g., parameter count, training time) relative to the baseline methods.**
>
> We thank the reviewer for pointing out this omission. Following their suggestion, we added a detailed comparison of computational overhead across methods in Figure 7 (Section B.6). All methods were evaluated under identical hardware settings for fairness.
>
> **Findings:**
> - C-DCWM incurs a higher per-step training cost, primarily due to the discrete latent world model and context conditioning.
> - However, C-DCWM converges in fewer training steps, which compensates for the per-step cost.
> - During evaluation, C-DCWM is slightly slower because the policy is conditioned on latent states obtained from the observation encoder.
>
> This analysis provides a clearer understanding of the computational trade-offs associated with C-DCWM. We thank the reviewer for helping us improve our manuscript.

---

> > ### Comment · Reviewer_iC8p · 2025-11-24
> >
> > Thanks for authors' response. The comprehensive experiments and explanations have basically addressed my concerns. The additional experiments provide a more thorough demonstration of C-DCWM's performance in terms of generalizability and computational complexity.
> > However, I share the same concern as other reviewers regarding the limited novelty of this work: it appears to be a straightforward extension of DCMPC applied to the multi-task setting of TD-MPC2, combined with a generic InfoNCE loss.
> > In light of the above, I will raise my rating to 4.

---

> ### Author Response · Authors · 2025-11-25
>
> We are glad that our additional experiments addressed your concerns, and we appreciate the updated score. However, we respectfully disagree with the assessment regarding the novelty of our work. We now elaborate on the settings and our contributions.
>
> First, we would like to clarify that we evaluate C-DCWM in *Offline Meta-RL* settings. While these settings share similarities with offline multi-task RL, they are fundamentally different and more challenging. In multi-task RL, agents are provided with explicit information about the task--such as a task ID or the true variation factors--during both training and testing. The objective is to learn a single policy that performs well across the same set of tasks by exploiting shared structure and representations. In contrast, in Meta-RL, **no prior information about the current task is provided at either training or test time**. The agent must *infer* the task efficiently from interaction data. The goal is to learn a policy capable of generalizing to previously unseen tasks.
>
> We acknowledge that C-DCWM borrows the discrete latent space design from DCWM \[1\] and that InfoNCE is widely used. However, we argue that **leveraging latent world models for Offline Meta-RL has not been explored in prior work**. Our contributions show that:
> - Latent world models conditioned on task representation (z) improve generalization to unseen tasks (Section 4.1).
> - World modeling, compared to pure reconstruction, better captures underlying task structure based on disentanglement and contrastive learning enhances task distinguishability (Section 4.3 and Section B.2).
> - Combining world modeling with contrastive learning leads to improved generalization on out-of-distribution tasks (Section B.3).
>
>
> **References**
>
> \[1\] Scannell, Aidan, et al. *Discrete Codebook World Models for Continuous Control.* The Thirteenth International Conference on Learning Representations. 2025.

---

### Official Review · Reviewer_2Hv7 · 2025-11-01

**Soundness:** 3
**Presentation:** 3
**Contribution:** 2
**Rating:** 4
**Confidence:** 3

**Summary:**

The paper presents Contextual Discrete Codebook World Models (C-DCWM) for offline meta-reinforcement learning. It jointly learns (i) a context encoder that infers a task representation z from short histories, and (ii) a latent world model conditioned on z, built with a finite scalar quantization (FSQ) module and trained via cross-entropy-based temporal consistency and contrastive InfoNCE objectives. The latent representations are then used to train policies and critics (IQL) in the quantized latent space. Experiments on MuJoCo, Contextual-DMC, and Meta-World benchmarks show improved in-distribution and out-of-distribution performance compared to several existing meta-RL baselines.

**Strengths:**

1. The integration of context-conditioned latent world models is clearly implemented and experimentally well-controlled.

2. Empirical performance across multiple offline meta-RL benchmarks is consistent, showing that classification-based temporal consistency can outperform standard regression losses.

**Weaknesses:**

1. Limited novelty. The approach mainly combines existing ingredients, e.g., context encoders from meta-RL, discrete latent world models from Dreamer/TD-MPC, and contrastive representation learning, into a joint training scheme. While the integration is clean, the conceptual advance over prior work such as CSRO, UNICORN, and discrete-latent Dreamer variants is marginal.

2. Overreliance on IQL head. The offline RL component is fixed to IQL; no evidence is provided that the representation benefits generalize across different offline learners (e.g., CQL, TD3+BC).

3. Missing comparison to contemporary discrete-latent models. Direct comparison to recent planning-based or discrete-latent world models (e.g., Dreamer-V3-Discrete, TD-MPC2 variants) is absent.

**Questions:**

None.

---

> ### Author Response · Authors · 2025-11-20
> **Response to Reviewer 2Hv7**
>
> We thank Reviewer 2Hv7 for their constructive comments and are pleased that the experimental setup was viewed as well-controlled and consistent. Their feedback directly contributed to strengthening the manuscript and improving several aspects of the evaluation.
>
> **Q1: Overreliance on IQL head. The offline RL component is fixed to IQL; no evidence is provided that the representation benefits generalize across different offline learners (e.g., CQL, TD3+BC).**
>
> We thank the reviewer for raising this point. In response, we expanded our empirical evaluation to include CQL and TD3+BC, reported in Table 14 (Section B.5). Their comment prompted us to verify whether our learned representations generalize across different offline RL learners.
>
> To ensure a fair comparison, we use default hyperparameters without per-environment tuning.
>
> **Findings:**
> - IQL remains the most robust across environments.
> - CQL performs competitively and even outperforms IQL in two environments, demonstrating that the benefits of our representations are not specific to IQL.
> - TD3+BC underperforms relative to both IQL and CQL, consistent with prior offline RL literature.
>
> We appreciate the reviewer’s suggestion, as adding these comparisons strengthened the empirical evidence that our representations are broadly useful.
>
> **Q2: Missing comparison to contemporary discrete-latent models. Direct comparison to recent planning-based or discrete-latent world models (e.g., Dreamer-V3-Discrete, TD-MPC2 variants) is absent.**
>
> We thank the reviewer for highlighting this gap. Based on their suggestion, we included a comparison to DreamerV3 [1] in Table 15 (Section B.7). This addition improved the completeness of our evaluation and clarified how C-DCWM relates to recent world-model approaches.
>
> For a fair zero-shot evaluation in the offline meta-RL setting, we reset DreamerV3’s RSSM hidden state at the initial timestep.
>
> **Key observations:**
> DreamerV3 struggles to generalize in tasks where optimal behavior varies substantially across tasks, such as:
> - Cheetah-speed, Finger-speed, and Walker-speed tasks (requiring different directional movements and speeds),
> - Ant-dir (requiring movement in different directions).
>
> It performs better in settings where optimal task-specific policies are more similar (e.g., Hopper-mass, Walker-friction). These results highlight the challenges faced by existing world models in offline meta-RL. We also note that, in the offline setting, DreamerV3 may exploit inaccuracies in poorly modeled regions due to the absence of an uncertainty penalty.
>
> Regarding TD-MPC2 [2], we thank the reviewer for this valuable suggestion. However, TD-MPC2 assumes access to task identifiers, which conflicts with the task-agnostic assumptions of offline meta-RL. While one could condition TD-MPC2 on our learned task embedding \(z\), doing so effectively constructs a new algorithm. In Section 4.2, we already analyze alternative latent formulations, and our SimNorm+MSE variant resembles a TD-MPC2–style latent structure.
>
> We thank the reviewer, as this feedback helped us clarify the conceptual relationship between our method and modern world models.
>
> ### References
> [1] Hafner, Danijar, et al. “Mastering diverse control tasks through world models.” *Nature* (2025).
> [2] Hansen, Nicklas, Hao Su, and Xiaolong Wang. “TD-MPC2: Scalable, Robust World Models for Continuous Control.” *International Conference on Learning Representations* (2024).

---

### Official Review · Reviewer_62LT · 2025-11-01

**Soundness:** 3
**Presentation:** 3
**Contribution:** 3
**Rating:** 6
**Confidence:** 4

**Summary:**

This paper proposes a novel offline meta-RL method called Contextual Discrete Codebook World Models (C-DCWM). C-DCWM encodes offline datasets from different tasks into task related representations. The world model is then conditioned on different learned representations. Experiments demonstrate that jointly training the world model and the context encoder leads to improved generalization performance.

**Strengths:**

1.The paper is clearly written and easy to follow.
2.The idea of extending DCWM to a conditioned version for generalizing across different tasks, and jointly training the model to obtain better task representations, is novel and well-motivated.
3.The authors have conducted comprehensive experiments and ablation studies to verify and analyze the effectiveness of C-DCWM.

**Weaknesses:**

1. While the current experimental validation is primarily conducted on continuous control tasks, these environments typically feature consistent dynamics within each task, which likely simplifies the learning of a discrete codebook. To further substantiate the generalizability of the proposed method, it would be compelling to evaluate it on more heterogeneous domains, such as the Atari benchmark. In these environments, a single task (or game) often comprises distinct levels requiring diverse policies and decision-making skills, presenting a more significant challenge for codebook learning. Extending the analysis to such tasks would greatly strengthen the empirical evidence for the method's robustness and effectiveness.
2. some figures and tables are placed far from where they are referenced. For example, Figure 1 is distant from Section 3, which makes it inconvenient to cross-reference the figure with the corresponding formulas and explanations.

**Questions:**

1.The placement of the figures and tables could be reorganized for better readability. For instance, Figure 1 might be placed after current Figure 2.
2. As I understand it, to generalize the policy to an out-of-distribution task, C-DCWM requires a dataset from that task to compute the task representation. However, this assumption may be unrealistic, as we may not always have access to a dataset for an unknown task. Would it be possible instead to compute $z$ in an online manner—for example, starting from an initial $z$ and updating the task representation after each interaction step?

---

> ### Author Response · Authors · 2025-11-20
> **Response to Reviewer 62LT**
>
> We thank Reviewer 62LT for their comments and are pleased that they found C-DCWM to be novel, well-motivated, and clearly presented. Their feedback helped us refine the manuscript’s clarity and improve the organization of our experimental discussion.
>
> **Q1: The placement of the figures and tables could be reorganized for better readability. For instance, Figure 1 might be placed after current Figure 2.**
>
> We appreciate the reviewer’s suggestion regarding figure placement. Following their recommendation, we reorganized the figures and tables so that each one appears closer to the section where it is discussed. This improved the readability and flow of the paper, and we thank the reviewer for pointing this out.
>
> **Q2: As I understand it, to generalize the policy to an out-of-distribution task, C-DCWM requires a dataset from that task to compute the task representation. However, this assumption may be unrealistic, as we may not always have access to a dataset for an unknown task.**
>
> We thank the reviewer for raising this important clarification point. Their question helped us explicitly describe how the task representation is handled during meta-testing.
>
> In the offline meta-RL setting, during meta-training, the task representation \(z\) is inferred from the offline datasets. However, during meta-testing, we do not assume access to a dataset from the unseen task. Instead, the task representation is:
>
> - initialized as \(z = 0\), and
> - updated online using interaction data collected by the agent.
>
> As the agent gathers experience, the task representation becomes increasingly accurate, enabling adaptation to new tasks without requiring an offline dataset. In response to this comment, we added an explicit paragraph in Section 4.1 to clarify this process.

---

### Author Response · Authors · 2025-11-20
**General Response**

We thank all reviewers for their constructive feedback. Your comments directly helped us strengthen the manuscript, broaden our empirical evaluation, and clarify several methodological aspects. Below, we highlight the additional experiments and analyses we conducted in response to your suggestions.

### Experiments and Additional Analysis
1. **Generalization to new environments (thanks to Reviewer iC8p).**
Reviewer iC8p’s comments encouraged us to expand our empirical study to more challenging cross-environment generalization settings. In response, we evaluated our method, C-DCWM, on the ML10 and ML45 settings in the Meta-World benchmark. These settings involve training on 10 or 45 environments and evaluating on five structurally related but unseen environments. C-DCWM, outperforms baselines on both training and test tasks across both benchmarks. Increasing the number of training environments slightly improves generalization; however, a notable gap remains between training and testing environments. These new results (Table 2 in the revised manuscript, Section 4.1) provide a clearer and more nuanced understanding of C-DCWM’s generalization capabilities.

2. **Effect of model size and number of parameters (thanks to Reviewer iC8p).**
Reviewer iC8p’s question about parameter count motivated us to perform a more extensive scaling analysis. We trained all methods across six different model sizes and compared performance. C-DCWM, which naturally uses more parameters due to its discrete latent model, shows more reliable performance scaling than the baselines (Figure 6, Section B.4). C-DCWM, with the default model size, outperforms baselines across all model sizes. However, at smaller parameter budgets, C-DCWM can underperform baselines because the policy and value networks must be reduced to accommodate the discrete world model. This experiment clarified that C-DCWM’s gains are not merely due to increased model capacity.

3. **Offline RL backbone comparison (thanks to Reviewer 2Hv7).**
Reviewer 2Hv7 raised an important concern regarding possible overreliance on Implicit Q-Learning (IQL). In response, we added experiments using CQL [1] and TD3+BC [2] as alternative offline RL backbones (Table 14, Section B.5). We find that CQL performs competitively with IQL and even outperforms it on certain environments, while TD3+BC performs worse overall. These results demonstrate that the benefits of C-DCWM extend beyond the choice of the offline RL component. We thank the reviewer for highlighting this point.

4. **Comparison to DreamerV3 (thanks to Reviewer 2Hv7).**
In response to Reviewer 2Hv7’s suggestion, we conducted a comparison between C-DCWM and DreamerV3 [3], a state-of-the-art model-based RL method with a discrete latent space. As shown in Table 15 (Section B.7), DreamerV3 struggles with zero-shot generalization in the offline meta-RL setting, particularly in environments requiring distinctly different optimal behaviors (e.g., forward vs. backward movement). This comparison, directly motivated by the reviewer’s feedback, helped contextualize C-DCWM within the world-model literature.

5. **Computational complexity (thanks to Reviewer iC8p).**
In response to the reviewer’s request, we added an analysis of computational cost (Figure 7, Section B.6). C-DCWM incurs a higher per-step training cost but converges in fewer iterations. At test time, inference is slightly slower due to the latent-space encoding. This analysis clarifies the trade-offs associated with discrete world models and strengthens the empirical discussion.

### References
[1] Kumar, Aviral, et al. “Conservative q-learning for offline reinforcement learning.” *Advances in Neural Information Processing Systems* 33 (2020).
[2] Fujimoto, Scott, and Shixiang Shane Gu. “A minimalist approach to offline reinforcement learning.” *Advances in Neural Information Processing Systems* 34 (2021).
[3] Hafner, Danijar, et al. “Mastering diverse control tasks through world models.” *Nature* (2025).

---

### Author Response · Authors · 2025-12-04
**Summary of the rebuttal phase for AC**

Dear appointed Area Chair,

Due to exceptional circumstances, we provide a summary of the discussions during the rebuttal phase and how we addressed the reviewers’ concerns. The reviewers raised issues regarding cross-environment generalization, model sizes, reliance on a specific offline RL algorithm (IQL), comparison to DreamerV3, computation cost, and novelty.
Below, we outline the additional experiments and analyses conducted in response. We respectfully ask the AC to consider these updates when making the final decision.

### Reviewer iC8p (Rating: 2)
The reviewer confirmed that the updated manuscript addresses their concerns.

1. **Generalization to new environments** (see `1` in the General Response and `Q1` in the specific response for details):
   We evaluated C-DCWM and the baselines on the ML10 and ML45 settings in the Meta-World benchmark. C-DCWM outperforms the baselines on both training and held-out testing tasks across both settings.

2. **Scaling the models** (see `2` in the General Response and `Q2` in the specific response for details):
   C-DCWM, with its default model size, outperforms all baselines across the full range of model capacities.

3. **Computational complexity** (see `5` in the General Response and `Q3` in the specific response for details):
   C-DCWM has a higher per-step training cost but converges in fewer iterations. At test time, inference is slightly slower due to latent-space encoding.

**Reviewer iC8p** increased the rating from **2 to 4** and stated:
> The comprehensive experiments and explanations have basically addressed my concerns. However, I share the same concern as other reviewers regarding the limited novelty of this work: it appears to be a straightforward extension of DCMPC applied to the multi-task setting of TD-MPC2, combined with a generic InfoNCE loss.

In response, we clarified that we evaluate C-DCWM in *Offline Meta-RL* settings, not *Offline Multi-Task RL*. We respectfully disagree regarding the novelty, as we show that:
- World modeling, compared to pure reconstruction, better captures the underlying task structure through disentanglement and consequently improves generalization to both in-distribution and out-of-distribution unseen tasks.

Please see our specific response for more details.

### Reviewer 2Hv7 (Rating: 4)
We addressed the reviewer’s concerns by conducting the additional experiments they requested.

1. **Alternative Offline RL backbones** (see `3` in the General Response and `Q1` in the specific response for details):
   We show that C-DCWM can effectively leverage different offline RL methods for policy optimization (we report results with CQL and TD3+BC in addition to IQL).

2. **Comparison to DreamerV3** (see `4` in the General Response and `Q2` in the specific response for details):
   DreamerV3 struggles with zero-shot generalization—particularly in environments requiring distinctly different optimal behaviors.

**Reviewer 2Hv7** did not react to our response. We believe that we have fully addressed the reviewer’s concerns and that, under normal conditions, they would have updated their rating.

### Reviewer 62LT (Rating: 6)

1. **Placement of figures and tables** (see `Q1` in the specific response for details):
   We reorganized figures and tables so that each appears closer to the section in which it is discussed.

2. **Computing task representations** (see `Q2` in the specific response for details):
   We clarified that the task representation is initialized as \(z = 0\) at meta-test time and is updated online using interaction data.

**Reviewer 62LT** also did not react to our response.

### Overall assessment
- All reviewers provided constructive feedback, which we addressed via additional experiments and clarifications.
- Reviewer `62LT` appeared to have a misunderstanding about how task representations are computed; we clarified this in Section 4.1.
- Reviewer `iC8p` assumed a multi-task evaluation; we clarified that our evaluation is in the Offline Meta-RL setting.
- We respectfully disagree with Reviewers `iC8p` and `2Hv7` regarding the characterization of novelty; we provide evidence in the manuscript showing that the contextual world model yields improved disentanglement and generalization beyond reconstruction-based approaches.

---

### Meta-Review · Area_Chair_jikc · 2025-12-30

**Summary:**

This paper proposes an offline meta-RL method called Contextual Discrete Codebook World Models (C-DCWM), with the key idea as follows: World models are conditioned on the task representation and trained jointly with the context encoder. More specifically,   C-DCWM encodes offline datasets from different tasks into task related representations; and then the world model is then conditioned on different representations learned from offline datasets.

**Reviewer Concerns:**

There are a few major concerns: 1) limited novelty: the reviewers see this work is a straightforward extension of the DCWM framework to the multi-task setting by conditioning it on a task representation, which I agree to.
2) Generalization to new tasks: the reviewers are generally concerned with the generalization of the proposed C-DCWM, especially in challenging and more heterogeneous domains. The authors did some more experimental studies to address this concern, but more work is definitely needed to get a better understanding.

**Reviewer Scores:**

The review scores are 6/4/2 (updated to 4).

---

### Decision · Program_Chairs · 2026-01-26

Reject